# Is Melatonin the “Next Vitamin D”?: A Review of Emerging Science, Clinical Uses, Safety, and Dietary Supplements

**DOI:** 10.3390/nu14193934

**Published:** 2022-09-22

**Authors:** Deanna M. Minich, Melanie Henning, Catherine Darley, Mona Fahoum, Corey B. Schuler, James Frame

**Affiliations:** 1Department of Human Nutrition and Functional Medicine, University of Western States, Portland, OR 97213, USA; 2Department of Sports and Performance Psychology, University of the Rockies, Denver, CO 80202, USA; 3College of Naturopathic Medicine, National University of Natural Medicine, Portland, OR 97201, USA; 4School of Naturopathic Medicine, Bastyr University, Kenmore, WA 98028, USA; 5School of Nutrition, Sonoran University of Health Sciences, Tempe, AZ 85282, USA; 6Department of Online Education, Northeast College of Health Sciences, Seneca Falls, NY 13148, USA; 7Natural Health International Pty., Ltd., Sydney, NSW 2000, Australia; 8Symphony Natural Health, Inc., West Valley City, UT 84119, USA

**Keywords:** melatonin, phytomelatonin, vitamin D, sleep, circadian rhythm, antioxidant, blue light, chronobiotic, chrononutrition, darkness deficiency

## Abstract

Melatonin has become a popular dietary supplement, most known as a chronobiotic, and for establishing healthy sleep. Research over the last decade into cancer, Alzheimer’s disease, multiple sclerosis, fertility, PCOS, and many other conditions, combined with the COVID-19 pandemic, has led to greater awareness of melatonin because of its ability to act as a potent antioxidant, immune-active agent, and mitochondrial regulator. There are distinct similarities between melatonin and vitamin D in the depth and breadth of their impact on health. Both act as hormones, affect multiple systems through their immune-modulating, anti-inflammatory functions, are found in the skin, and are responsive to sunlight and darkness. In fact, there may be similarities between the widespread concern about vitamin D deficiency as a “sunlight deficiency” and reduced melatonin secretion as a result of “darkness deficiency” from overexposure to artificial blue light. The trend toward greater use of melatonin supplements has resulted in concern about its safety, especially higher doses, long-term use, and application in certain populations (e.g., children). This review aims to evaluate the recent data on melatonin’s mechanisms, its clinical uses beyond sleep, safety concerns, and a thorough summary of therapeutic considerations concerning dietary supplementation, including the different formats available (animal, synthetic, and phytomelatonin), dosing, timing, contraindications, and nutrient combinations.

## 1. Introduction

Due to the interest in immune health prompted by the pandemic and its lasting effects on mental health and sleep disturbances, melatonin (*N*-acetyl-5-methoxytryptamine) has become a popular topic of media discussion as well as a research interest, with several publications steadily rising every year. In conjunction with the plethora of scientific discoveries, there has been heightened awareness by consumers and health practitioners for its use as a sleep aid and immune health, most likely resulting in the more than doubling of melatonin dietary supplement sales in the U.S. to USD 821 million in 2020 compared with 2017 [1].

With its surge in popularity, concern about low levels, and correspondingly high dietary supplement sales, it has been generally suggested that melatonin is the “next vitamin D.” Aside from the trends, there are several scientific reasons for this comparison. From a basic mechanistic perspective, both have widespread effects on multiple systems and are found extensively throughout the body. While their multi-functionality may seem beneficial and broadly applicable to disease states, the possible risk is identifying with them as panaceas. As with any nutrient, there can be a spectrum of deficiency, excess, toxicity, or imbalance. Like vitamin D, melatonin could plausibly be treated as a required dietary nutrient, with certain individuals necessitating less, more, or fine-tuning to personalized needs for lifestyle requirements (e.g., exposure to artificial blue light at night, jet lag, shift work) or alterations in physiology, such as gene variants in melatonin receptors or metabolic response [2,3].

In addition to being compared with vitamin D, melatonin has also been referred to as “nature’s most versatile biological signal” [4] since its clinical application surpasses sleep. The classification of melatonin has been wide-ranging, from pineal hormone to amphiphilic antioxidant. It is a ubiquitous molecule, an indoleamine, produced endogenously in animals and plants. As a result, humans continually either ingest it from exogenous dietary sources or produce it endogenously. In humans, it is largely produced from the amino acid tryptophan by the pineal gland and in the gut-residing enterochromaffin cells. Even though the pineal gland receives much attention for its melatonin production, there is 400 times more melatonin in the gut mucosa [5].

On average, the pineal gland produces between 0.1 and 0.9 mg of melatonin per day [4,6]. Melatonin production and circadian rhythms do not develop in babies until around three months [7]. Breastfed babies have the benefit of melatonin from their mother’s milk [8]. Levels from infancy to adolescence increase and plateau in association with Tanner stages of puberty and then slowly decline with age starting in the late twenties [9,10]. Children typically produce more melatonin than adults, which may infer that their need for dietary supplementation may require further scrutiny and be limited to specific disease states [4,11]. Production gradually declines as people age, starting in the late twenties to the fifties, with production leveling at approximately 30 pg/mL [4,11] (see Figure 1).

Aside from aging, production of melatonin can be influenced by illness [12], diet [4], environmental factors like bright light at night [13], medication use [14], and lifestyle [15]. Of interest, research has indicated that the sheer amplitude of plasma melatonin may not have as much to do with chronological age but more with the degree of pineal calcification and associated melatonin secretion [16]. However, that perspective begs the question of why the pineal gland becomes calcified and how it may be decalcified [16]. In our modern era, perhaps the largest contributor to melatonin imbalance would be those subject to jet lag, shift work, overuse of artificial light at night (e.g., from cell phones, computers, and fluorescent/LED light), or challenges to their circadian rhythm due to environmental or seasonal changes.

Melatonin is colloquially referred to as the “hormone of darkness” since it is produced in response to darkness, as perceived by the eye’s retina [4]. Its synthesis is reduced by exposure to light, with artificial light reducing a person’s melatonin production and increasing disease risk [4,6]. From a practical and even clinical perspective, vitamin D and melatonin may act as biochemical sensors to meet requirements for both light and darkness, respectively. Thus, it might be loosely implied that vitamin D deficiency indicates a “sunlight deficiency” in perhaps the same manner that melatonin secretion could be affected by a “darkness deficiency,” where there is overexposure to artificial, blue light at night, disabling the signal to the pineal gland to produce it for initiating sleep (See Figure 2). There may even be levels of crosstalk and overlap between them that have not yet been fully elucidated but might have clinical relevance. For example, it has been demonstrated that melatonin can bind several target proteins, including enzymes, receptors, pores, and transporters [17]. Most relevant for the discussion of this paper is that it can bind the vitamin D receptor (VDR), resulting in an enhancement of vitamin D’s signaling effects and subsequent cellular activities [18].

Like vitamin D, melatonin is found throughout the body. Melatonin has been found in many tissues other than the pineal gland and gut mucosa, including the brain, retina, lens, cochlea, trachea, skin, liver, kidney, thyroid, pancreas, thymus, spleen, and reproductive tissues [6]. It is present in nearly all bodily fluids: cerebrospinal fluid, saliva, bile, synovial fluid, amniotic fluid, urine, feces, semen, and breast milk [16,19,20]. Specifically, vitamin D and melatonin may work synergistically in the skin. Ultraviolet (UV)-B radiation is required to convert 7-dehydrocholesterol in the skin to vitamin D3 [21]. At the same time, melatonin is an antioxidant in the skin to ward off the damaging effects of UV light. In the future, there may be more skincare innovations that involve both vitamin D and melatonin due to the activities they share in the skin [22].

As an adjunct to its well-known role in sleep, melatonin has been seen as a prominent cellular guard against oxidative stress, specifically linked to the redox status of cells and tissues. In fact, it has been suggested to be one of the most potent antioxidants because of its ability to scavenge up to 10 reactive oxygen (ROS) and nitrogen species (RNS) with its metabolites compared with most antioxidants, which may only be able to quench a few ROS [16,23,24]. Finally, melatonin is involved with multiple activities that include mitochondrial homeostasis, genomic regulation, modulation of inflammatory and immune cytokines, directly impacting both systemic and acute anti-inflammatory properties as well as indications around its potential role in phase separation [25,26]. It has been proposed that both vitamin D and melatonin orchestrate many of their functions, especially related to redox status, at the level of the mitochondria [27]. Concurrent with the age-related depletions in levels of vitamin D and melatonin, there is mitochondrial dysfunction, which has implications in a variety of clinical conditions that present differently through the seasons with changing light exposure [27].

In this review article, the current clinical uses for melatonin based on body systems and functions will be assessed. Therapeutic considerations, dietary sources, types of dietary supplements, and dosing and timing of supplementation are key aspects of the clinical discussion. Newer, in vitro data on phytomelatonin would suggest superior antioxidant, anti-radical, and anti-inflammatory effects over synthetic melatonin. Furthermore, any safety concerns, such as those associated with the synthesis of melatonin or supplemental formats, will be elaborated upon herein. In several ways, based on melatonin’s mechanisms of action and clinical applications, there are some relative similarities and complementary activities between vitamin D and melatonin that are worthwhile to note throughout the discussion (see Table 1).

## 2. Scientific Mechanisms

Melatonin is a versatile molecule produced in multiple parts of the body. It has long been known to be a potent antioxidant and anti-inflammatory agent, and several published review articles have already detailed these aspects of melatonin. Therefore, in this section, newer research on its mechanism of action will be discussed, providing the necessary foundational information to understand its (potential) application to clinical conditions.

### 2.1. Mechanisms Related to Aging and Disease: Antioxidant Defense, Oxidative Stress Reduction, and Anti-Inflammatory Properties

Numerous studies have identified melatonin as a powerful free-radical scavenger with potential protective properties against neurodegenerative disorders, epilepsy, and certain cancers. Recent in vitro and in vivo studies have continued to fortify the foundational aspects established over the past decades that would attest to melatonin’s roles in antioxidant defense, oxidative stress reduction, and anti-inflammatory processes [29,30]. Notably, as stated above, it is a highly efficient antioxidant as one molecule of melatonin can scavenge multiple (~10) reactive oxygen and nitrogen species through a cascading mechanism related to its secondary, tertiary, and even quaternary metabolites [24,31]. In addition, melatonin has the dual capability of targeting receptor-independent and receptor-dependent processes. These studies further report that melatonin has significant abilities to block pro-inflammatory processes acting on cyclooxygenase (COX-2) and enhance programmed cell death (apoptosis) in aberrant cells [29,32], which would theoretically make it a desirable therapeutic in diseases of aging (“inflammaging”), such as cancer. Melatonin’s dual actions can inhibit pro-oxidative enzymes (e.g., xanthine oxidase) while also acting to potentiate the critical antioxidant enzymes such as superoxide dismutase (SOD), glutathione peroxidase, and catalase, aiding in the body’s first line of immune defense and metabolic detoxification [31,32]. Overall, age-related decreases in endogenous melatonin production are correlated with disease and dysfunction. In vitro and in vivo studies demonstrate melatonin’s protective ability against mitochondria-mediated injury with hypertension and obesity, suggesting that dietary supplementation with exogenous melatonin in later years may be an effective therapeutic intervention for such age-related conditions [33].

Kukula-Koch et al. [34] performed cellular assays to determine if superior effects in anti-radical, antioxidant, and anti-inflammatory activities exist in phytomelatonin compared to the synthetic form. Based on these in vitro results using human cell lines, they reported significant benefits with phytomelatonin compared to synthetic melatonin [34]. Phytomelatonin was found to have 646% stronger COX-2 inhibition (see Figure 3), 267–470% more potent free-radical scavenging ability (see Figure 4), and 100% greater efficacy in reducing cellular ROS in a human skin cell line (see Figure 5) when compared to synthetic melatonin, most likely due to the other constituents found in phytomelatonin such as chlorophyll, beta-carotene, lutein, and other protective, antioxidant phytonutrients [values derived from the original data presented in [34].

### 2.2. The Central Role of the Mitochondria

Newer data indicate that the mitochondria are pivotal for several aspects of melatonin: its production, metabolism, and activity through receptors [35]. Rather than respond to the signals of the light/dark cycle or the pineal gland, the mitochondria can induce the production of melatonin based on intracellular need. Levels of melatonin are known to be higher in mitochondria compared with blood levels, most likely due to the greater antioxidant requirements with the copious amounts of free radicals generated through the electron transport chain [25]. Melatonin assists in mitochondrial redox balance through its ability to reduce the superoxide anion molecule from the electron transport chain and to directly scavenge free radicals [36]. In addition to these functions, melatonin facilitates mitochondrial function by encouraging healthy endogenous levels of antioxidant defense enzymes like superoxide dismutase.

Mitochondrial dysfunction is one of the mechanisms related to diseases of aging [36]. It may be that declining melatonin levels with age and, thus, less protection of the mitochondria from oxidative stress may be contributors to preclinical changes and, ultimately clinical symptoms. There may be the ability to offset some of the decline accompanying the aging process through physiological regeneration with supplemental melatonin, as has been documented in animal and human studies [36].

### 2.3. Gut-Synthesized Melatonin the Gut Microbiome

Two types of cells are responsible for the production of melatonin: pinealocytes and enterochromaffin cells. Pinealocytes are located in the pineal gland within the brain. Enterochromaffin cells are located on the surface of the entire gastrointestinal (GI) tract, with high concentrations in the mucosal lining of the GI tract. Pinealocytes are affected by light and dark; exposure to light suppresses melatonin production and release from the pinealocytes, while darkness (when registered by the retina) increases melatonin production and release into the bloodstream starting with vessels in the brain. From the blood vessels in the brain, melatonin is carried to other body tissues. It is estimated that the enterochromaffin cells within the gut contain upwards of 400 times the amount of melatonin than what is produced by pinealocytes. Levels of gut melatonin can be anywhere from 10 to 100 times greater than melatonin in blood serum levels [37,38].

Unlike pinealocytes, enterochromaffin cells are not regulated by light and dark but appear affected by food intake and digestion [39,40]. Of note, it remains speculative as to how pineal-produced and gut-derived melatonin interrelate, whether there is any gut-pineal axis crosstalk, and how different dietary patterns or even specific foods, fasting regimens, or timing of meals (chrononutrition) may alter systemic melatonin levels or the physiological relevance of any changes. This area of diet and gut-produced melatonin is rich with questions to be answered through research.

The release of melatonin in the gut acts in a paracrine manner to increase activity and circulation in the gastric mucosa and enhances GI motility [5,41]. With the increased production of gastrin, melatonin has also been attributed to increasing the tone of the lower esophageal sphincter [42]. Further, melatonin has anti-excitatory properties in the gut. It can stimulate the regeneration of epithelial cells [43] and has also been shown to have protective antioxidant effects on the lining of the GI tract [43].

An investigation into gut microbiota has identified microbial influences on the serotonergic and melatonergic systems [25]. The established serotonergic and melatonergic systems may be vulnerable before establishing a stable global biota in infancy. Elderly individuals may also be more susceptible to serotonergic and melatonergic system errors due to known lower amounts of biota diversity. Both serotonergic and melatonergic systems are also prone to immune and inflammatory responses, adding to the complexities of the gut-brain axis [44].

It is suspected that the gut-brain axis is a network of complex interactions between the nervous and GI systems with major contributions from intestinal microbiota. Ultimately, the gut microbiota may influence CNS function and, over time, could cause neurological diseases, including Alzheimer’s disease, mood and anxiety disorders, multiple sclerosis (MS), Parkinson’s disease, and migraines. Preliminary animal research indicates a relationship between gut dysbiosis, endogenous melatonin production, and pathological changes associated with Alzheimer’s disease [37]. Along these lines of research, melatonin administration in animals helped reduce dysbiosis due to sleep restriction [45,46]. In one of the studies, *Akkermansia muciniphila* and *Lactobacilli* species were increased in the melatonin-treated animals [46].

#### Gut Health, Dietary Polyphenols Melatonin

While still in exploratory phases, there may be eventual complementary action between melatonin and polyphenols. Although not absorbed in the GI tract to a great extent, polyphenols have become highlighted for their impacts on gut health, particularly due to the secondary metabolites that form from their interaction with the microbiota [47]. There are initial indications that the antioxidant and anti-inflammatory actions of melatonin may work together with select polyphenols (e.g., resveratrol, epigallocatechin 3-gallate) for therapeutic indications [48,49,50,51,52,53,54].

### 2.4. Kynurenine Pathway, Energy Regulation Stress Response

The broader picture of mental health, specifically depression, has been correlated with melatonin levels. While low serotonin levels are known to be consistent with clinical depression, low melatonin levels also appear to have a significant connection. Low levels of melatonin may trigger an upregulation in the kynurenine pathway, and kynurenine production, as well as trigger the aryl hydrocarbon receptor (AhR) located on the outer membrane of the mitochondria [32,55,56,57,58]. The AhR is responsible for modulating mitochondrial metabolism, melatonergic pathways, acetyl-coenzyme A, and COX-2 prostaglandin. When AhR is triggered, overall endogenous pineal melatonin production is suppressed [32,55,56,57,58].

Melatonin is derived from serotonin using 5-hydroxytryptophan (5-HTP) and tryptophan [32,55]. Tryptophan is involved primarily in the melatonin-serotonin pathway and the kynurenic pathway. The melatonin-serotonin pathway accounts for approximately five percent of dietary tryptophan degradation, while the kynurenic pathway accounts for approximately 95 percent of dietary tryptophan degradation [32,56]. The kynurenic pathway is an essential process needed to convert tryptophan into nicotinamide adenine dinucleotide (NAD+) for cellular energy. Although the bisecting pathways of tryptophan are involved in separate processes, one pathway is thought to impact the other. Exercise has been found to increase the throughput in the serotonin-melatonin pathway, eventually increasing both serotonin and melatonin levels and impacting mood and cognition [55]. Conversely, acute or chronic inflammation and stress have been found to increase the throughput of the kynurenine pathway, leading to an increase in tryptophan’s conversion to kynurenine [32,55]. Kynurenine is a byproduct or metabolite produced when tryptophan is converted to niacin. High concentration levels of kynurenine in the brain are present in instances of depression [32,55]. Kynurenine is then converted to either kynurenic acid or quinolinic acid. Quinolinic acid is a neurotoxin, while kynurenic acid has neuroprotective properties [32,55,56,57,58].

In addition to the kynurenine pathway, Fila et al. [59] reported that the GI tract is also a major site for tryptophan metabolism. These pathways are interconnected, and when there is dysregulation with one, there is most likely dysregulation with the other. Many neurological disorders feature elements of the kynurenine pathway of tryptophan, specifically the dysregulation of tryptophan metabolism and subsequent melatonin production.

## 3. Clinical Uses

In this section, the latest clinical research on various health aspects and implications of chronic disease will be presented (see Table 2 for a summary).

### 3.1. Central Nervous System

Endogenous melatonin is produced from tryptophan by 5-HTP and serotonin. The bi-directional neural processes from the gut and the brain rely on specific metabolic reactions. These metabolic reactions are reliant on the conversion of tryptophan into serotonin. In many ways, serotonin provides the foundation for the connection between the gut-brain axis, as it directly affects and influences neurological response and central nervous system transmission. Tryptophan metabolism is directly influenced by inflammatory and immune responses, which trigger the throughput in the aforementioned kynurenine pathway.

Melatonin is both water- and lipid-soluble (‘amphiphilic’); thereby, it can freely flow among all bodily tissues, especially across the selective blood–brain barrier, making it likely one of the most formidable antioxidants within the central nervous system [60]. Preliminary research also indicates that it may be an active component in the glymphatic fluid, assisting in removing metabolic waste such as amyloid buildup [61]. Theoretically, based on this finding, it may be worthwhile from a therapeutic perspective to dose melatonin so that older adults with neurodegenerative conditions could increase cerebrospinal and glymphatic fluid levels. However, this concept is still in its infancy.

Neurodegenerative conditions share mitochondrial dysfunction in their pathogenesis. Mitochondria, the cellular energy source, are also the target of oxidative damage. The sensitive nature of mitochondrial membranes, which can be damaged by many factors, may find protection with the oral administration of melatonin [62]. Mitochondrial membranes selectively take up melatonin, a function not shared by other antioxidants [63].

#### 3.1.1. Circadian Rhythm Modulation

Human circadian rhythms are entrained to the environmental day primarily by light exposure, particularly at dawn and dusk [64]. Melatonin supplementation can also modulate the circadian rhythm by causing an advance or delay, depending on the administration time. In this manner, melatonin acts as a chronobiotic. The melatonin phase response curve specifies how exogenous melatonin will shift the individuals’ body clock when given at various times in relation to their sleep midpoint [65]. For instance, 0.5 mg and 3.0 mg of melatonin taken eleven hours before the sleep midpoint will cause a phase advance, so the individual feels sleepy earlier and awakens earlier. Both doses of melatonin taken in the morning, approximately 6 h after the sleep midpoint, will cause a phase delay. Of note, when melatonin is taken 4 h before the sleep midpoint, i.e., at bedtime, the low dose of 0.5 mg does not shift the circadian rhythm, while the 3.0 mg dose will cause a phase delay. This dosing regimen may contribute to the occasional complaints of a paradoxical effect of melatonin supplements on sleep.

##### Circadian Rhythm Sleep-Wake Disorders

Circadian rhythm sleep-wake disorders are either intrinsic or extrinsic. Intrinsic circadian rhythm disorders appear when the individual’s body clock is either off-set from the norm, as in delayed or advanced sleep-wake phase disorder, or irregular, as in non-24 sleep-wake rhythm disorder. Lifestyle factors can result in extrinsic circadian rhythm disorders such as shift work disorder or jet lag disorder [66]. Melatonin supplements can be used therapeutically for both conditions [67].

Studies using melatonin have explored how shift work, particularly night work with its exposure to light at night, may increase the risk of cancer, aggravate both gastrointestinal and cardiovascular disease, complicate pregnancy, and interfere with drug therapy [68]. Multiple studies, opinions, and guidelines have suggested melatonin as primary therapy for improved health and sleep of shift workers [69,70,71]. Thus, at a larger, more macroscopic level, as mentioned previously, imbalances in melatonin may be associated with what might be referred to as a “darkness deficiency,” or a lack of adequate evening darkness to initiate the secretion of melatonin by the pineal gland.

Delayed sleep-wake phase disorder is a persistent shift in sleep-wake times later than social norms, causing insomnia-like symptoms, difficulty waking in the morning, and excessive daytime sleepiness. This condition is best treated with precisely-timed melatonin, considering the desired bedtime and wake time. A randomized study of people with delayed sleep-wake phase disorder found that a low dose of melatonin (0.5 mg) an hour before the desired bedtime, along with behavioral strategies for four weeks, resulted in earlier sleep onset, improved sleep efficiency during the first third of the night, and reduced subjective complaints [72].

##### Jet Lag

With increasing travel and global connectivity, more individuals need to recover from jet lag sooner and faster. Many studies support melatonin’s use in reducing the ill effects of jet lag and speeding up the normalization of circadian rhythms [73]. In a Cochrane review, nine out of ten trials found that melatonin effectively reduced jet lag symptoms in travelers, especially if traveling eastward or over five time zones [74]. Specific phase-shifting protocols support the sleep phase during travel across time zones. Therapeutics include precisely-timed melatonin, light, and dark. These protocols are best known by sleep specialists [75].

##### Sleep Dysfunction

Although melatonin has, in some ways, become synonymous with sleep, other clinical approaches would serve as first-line interventions before using melatonin. Dysfunctional sleep does not have one mechanism, such as reduced melatonin, but potentially several causes, some or all, may be impacting melatonin levels. With over eighty sleep disorders [76], full assessment and diagnosis are important for effective treatment. Underlying inflammation-related diseases may need to be addressed, such as metabolic syndrome, sleep apnea, and any type of joint or muscle pain [77,78]. Even hormonal fluxes related to estrogen, cortisol, and insulin are essential to assess for imbalance and correct accordingly [79,80,81]. Moreover, there may be an association between environmental toxins such as heavy metals (e.g., arsenic) and sleep disturbance [82,83]. Sleep hygiene, such as room temperature, adequate darkness, noise, and comfort of bed and pillows, would be simple actions to ensure a healthy environment [84,85]. Further to the sleeping room, engaging in healthy lifestyle practices such as refraining from stimulants, eating or being on devices too close to bedtime, and unwinding from the day’s stresses with relaxation practices such as a warm bath or physical activity, need consideration [86,87]. From a nutritional standpoint, assessing dietary intake of macronutrients, especially tryptophan-containing sources of protein [87], along with micronutrients such as magnesium [88], vitamin D, and calcium, would be essential for ensuring a biochemical foundation that would foster healthy sleep [89]. Thus, melatonin would be utilized preferentially when the other changes have been implemented if there was an indication.

Melatonin has a hypothermic action. A decrease in core body temperature is soporific [90]. In this way, exogenous melatonin can have a direct effect on sleep. A meta-analysis of melatonin for the treatment of primary sleep disorders analyzed nineteen studies involving 1683 individuals. Melatonin had a statistically significant effect on reducing sleep latency and increasing total sleep time. Trials that used higher doses of melatonin and conducted over a longer duration demonstrated even greater effects on these two sleep issues, and overall sleep quality was also significantly improved in melatonin users [1,91].

A 2017 systematic review [92] identified 5030 studies on melatonin and sleep, but only twelve were included to meet their criteria for randomized, controlled, and single or double-blind studies. The summary concluded that melatonin is indicated for the following [92]:Insomnia: Immediate release 1–3 mg, 30 min before bed; slow-release can be used for sleep maintenance problemsRegulating sleep in blind individuals who often experience non-24 h sleep-wake rhythm disorder;Replicating the normal endogenous pattern;Delayed sleep phase.

In 2022, researchers at Harvard Medical School and Brigham and Women’s Hospital explored low (0.3 mg) and high (5 mg) dose melatonin in a relatively small sample of healthy older adults. Both doses improved sleep efficiency, but the higher dose impacted the biological day and night sleep patterns, the duration of non-REM sleep, and awakening time [93].

Melatonin supplementation has shown promise in a rare sleep disorder, idiopathic REM Behavior Disorder (iRBD). Patients with iRBD will retain muscle tone during REM sleep and can act out their dreams, posing a danger to themselves and others. Importantly, iRBD is a prodrome biomarker for Parkinson’s disease. In one study, six months of low-dose 2 mg melatonin supplementation taken at the same clock time (between 10–11 pm, personalized to the individual’s chronotype) resulted in a decrease in iRBD symptom severity over the first four weeks of treatment. This improvement was maintained over the follow-up period of 4.2 ± 3.1 years [94].

#### 3.1.2. Eye Health

With the retina as the target tissue perceiving light and signaling to the pineal gland, it is of interest to determine the role of melatonin in the retinal-pineal gland axis and related dysfunctions. In fact, individuals who are blind tend to have increased abnormalities in circadian rhythm compared with those who have sight [95]. Relatively smaller amounts of melatonin are produced in the retina compared with the pineal gland [96]. Despite the logical interrelationship between the retina photoreceptors and light sensitivity, there has not yet been a deep exploration into the utilization of melatonin for eye disorders. However, interest has been expressed for inflammatory conditions such as ocular neuritis and uveitis [97]. Some researchers suggest that glaucoma may be a therapeutic target for melatonin [98,99]. Age-related macular degeneration is another serious ophthalmic condition that theoretically could benefit from melatonin administration, although significant clinical research is currently lacking [100,101].

#### 3.1.3. Cognitive Conditions (Dementia)

Overall, clinical data suggest that melatonin supplementation improves sleep and neurotransmission and reduces sundowning in those with Alzheimer’s disease. At a mechanistic level, it may decrease the progression of the disease through its protection of neuronal cells from amyloid-beta, possibly due to the facilitation of its degradation and transport from the brain matter through the glymphatic fluid [102,103,104]. In a small pilot study of elderly patients with a mild cognitive deficit, the ability to remember previously learned items improved, and depression decreased with melatonin [105]. A more extensive, longer-term study found that patients with mild cognitive impairment scored better on the Mini-Mental Status Exam and the Sleep Disorders Index when taking melatonin [106]. Oxidative stress is one of the leading causes of age-related brain dysfunction by impairing neurogenesis. Thus, researchers are exploring influences on monoamine synthesis, a common target for diseases of the aging brain [107,108], as well as the potential of melatonin as a therapeutic in dementia.

#### 3.1.4. Migraines and Headaches

Migraines have a solid correlation to altered gut microbiota involving amines and indoles. Depleted gut melatonin may be involved in migraine occurrence because of the relative increase in *N*-acetylserotonin to melatonin ratios, resulting in hyperactive glutamatergic excitatory transmission in migraines. Migraines can also be correlated with many autoimmune disorders tied to melatonin regulation failure. These conditions include Hashimoto’s thyroiditis with associated hypothyroidism, rheumatic diseases, and antiphospholipid syndrome [109]. Ultimately, the gut microbiota may influence CNS function and, over time, could cause neurological diseases, including Alzheimer’s disease, mood and anxiety disorders, multiple sclerosis (MS), Parkinson’s disease, and migraines.

Migraine headaches are comorbid with several health conditions, including neurological, psychiatric, cardiovascular, cerebrovascular, GI, metaboloendocrine, and immunological disorders. It is suspected that the gut-brain axis is a network of complex interactions between the nervous and GI systems with significant contributions from intestinal microbiota. Many neurological disorders feature elements of the kynurenine pathway of tryptophan, specifically the dysregulation of tryptophan metabolism and subsequent melatonin production [109].

A randomized, multi-center, parallel-group design was conducted in which melatonin was compared with amitriptyline and placebo for twelve weeks. A 3 mg dose of melatonin reduced migraine frequency, demonstrating the same effectiveness as amitriptyline in the primary endpoint of the frequency of migraine headaches per month [109]. Melatonin was superior to amitriptyline in the percentage of patients with a greater than 50% reduction in migraine frequency, and melatonin was better tolerated than amitriptyline. It has also been reported as an effective treatment for primary headache disorders [109].

An additional surveillance study observed sixty-one patients diagnosed with chronic tension headaches [110,111]. Patients were given 3 mg of melatonin for thirty days following a baseline period and followed up after sixty days. Quality scores were obtained using VAS pain intensity, Hamilton Anxiety Rating Scale (HAM-A), and Hamilton Depression Rating Scale (HAM-D) at the study’s inception, post-thirty days of treatment, and post-sixty days of treatments. Overall, significant decreases in pain and tension headache-associated symptoms were observed after melatonin use. Sleep quality was also significantly improved during and after the study [110,111].

#### 3.1.5. Tinnitus

Melatonin has been used to treat chronic tinnitus in adults. One study observed a significantly greater decrease in tinnitus scores on an audiometric test and self-rated tinnitus after treatment with melatonin compared to placebo [112]. Hormonal influences such as puberty, the menstrual cycle, pregnancy, hormonal birth control, hormone replacement therapy, and menopause are possible explanations for why women may experience tinnitus. Other changes that could influence and worsen tinnitus during these times could be lack of sleep, fatigue, and stress.

#### 3.1.6. Attention-Deficit Hyperactivity Disorder (ADHD) and Autism

When it comes to attentional disorders and the autistic spectrum, the profound effects of melatonin may be far-reaching. Research groups have evaluated the genes that encode melatonin metabolism in patients with attention deficit hyperactivity disorder (ADHD) compared to controls. Genetic results suggest a melatonin-signaling deficiency in ADHD [113]. Sleep disorders are comorbid in those with ADHD, affecting cognitive, behavioral, and physical development. In most individuals with ADHD, there is a delayed circadian phase (evening preference) and subsequent issues with daytime function. In these individuals, endogenous pineal melatonin is significantly dampened during the evening hours (triggered by dim light).

Evidence suggests somewhat variable responses to supplemental melatonin in clinical ADHD. This variability could be due to differing or overlapping etiologies of ADHD, whether it is a manifestation of genetic SNPs related to sleep disturbance and circadian rhythm dysfunction or attributed to the melatonin-signaling deficiency. More research is needed to determine appropriate dosage protocols specific to the pathophysiology of ADHD under the supervision of a qualified healthcare professional [114,115].

Sleep disturbances in autism have led researchers to investigate melatonin’s role in this spectrum of disorders. It was found that autistic patients have low melatonin levels caused by a primary deficit in ASMT gene activity [116]. In a double-blind, placebo-controlled study, investigators tested children diagnosed with autism spectrum disorder (ASD) (*n* = 103) and healthy children (*n* = 73) for serum melatonin, the oxidants of nitric oxide, and malondialdehyde levels. Overall, children diagnosed with ASD and positive family history had higher serum melatonin and nitric oxide levels, with significantly lower malondialdehyde/melatonin ratios, suggesting greater impaired oxidant-antioxidant metabolism and balance in children with ASD [117].

A review article found that patients with autism had improved sleep parameters, better daytime behavior, and minimal side effects with melatonin use [118]. Research has suggested that melatonin is effective as a sleep inductor; doses between 1–5 mg can be used thirty minutes before bedtime. For delayed sleep phase syndrome, doses between 0.2–0.5 mg have been most effective when given six to eight hours before desired sleep [72].

### 3.2. Cardiometabolic Health

Improvements in LDL cholesterol and blood pressure have been shown in as few as two months of melatonin use (5 mg/day, two hours before bedtime) in thirty patients with documented metabolic syndrome who had not responded to a three-month intervention of therapeutic lifestyle modifications [119]. Further, melatonin has been shown to decrease nocturnal hypertension, improve systolic and diastolic blood pressure, reduce the pulsatility index in the internal carotid artery, decrease platelet aggregation, and reduce serum catecholamine levels [120,121,122,123]. A recent meta-analysis and systematic review by researchers at The Chinese University in Hong Kong concluded that a controlled-release oral melatonin supplement reduced asleep systolic blood pressure by 3.57 mm Hg [120].

Cai et al. [124] correlated low levels of endogenous melatonin to decreased long-term survival in patients with pulmonary hypertension. As illustrated, multiple mechanisms are involved with the pleiotropic abilities of melatonin that not only have been shown to have antioxidant, inhibition of oxidative stress, and anti-inflammatory effects but also in inducing vasodilation, cardio-protective, cancer-protective, and benefits in respiratory diseases. Melatonin levels were attributed to hyper-activation of the sympathetic system and/or the renin-angiotensin system in patients with pulmonary hypertension [124].

Other studies have shown that melatonin improves outcomes in patients with heart failure and is considered a preventive and adjunctive curative measure in these patients [123]. A randomized double-blinded placebo-controlled clinical trial with two parallel arms using either placebo or oral 10 mg melatonin supplementation per day for twenty-four weeks in patients with heart failure and reduced ejection fraction observed improvements in endothelial function in those who did not also have diabetes [125].

There has been some discussion as to whether melatonin may be helpful in conditions involving glycemic control, such as in non-insulin-dependent type 2 diabetes. A recent, relatively small, placebo-controlled study in male diabetics showed reduced insulin sensitivity by 12% after 10 mg of melatonin for three months [126]. The difference in effects of melatonin on oral glucose tolerance in the diabetic population may involve polymorphisms in the type 2 diabetes-associated G allele in the melatonin receptor-1B gene (MTNR1B) [3,127,128]. In one clinical trial with Spanish type 2 diabetics [129], the relationship between endogenous melatonin, dietary carbohydrate, and the effects of late-night eating were investigated. It was found that glucose tolerance was impaired in the late versus the early eating condition, especially in MTNR1B G-risk allele carriers, known to have insulin secretion defects. While this type of genotype is not easily assessed through current clinical laboratory assessment, it is best to monitor melatonin supplementation and any changes in blood sugar response in patients with glycemic control issues.

### 3.3. Reproductive Health

The hypothalamic-pituitary-gonadal axis is controlled through hormones. Although the data need further delineation, there is a bidirectional relationship between melatonin and sex steroids, especially estrogen, that is determined by many factors [130]. For example, the role of melatonin becomes increasingly important in the menopausal transition with sleep disturbances and changes in metabolism. Therefore, its effects on female health aspects such as pregnancy, fertility, and ovarian and uterine dysfunctions are worth noting in the sections below, highlighting relevant research in these areas.

#### 3.3.1. Pregnancy and Fertility

A review of the available literature by obstetric researchers found that because pregnancy has increased oxygen demands on the body and, thus, more free radical damage, melatonin supplementation may be a critical consideration for both complicated and normal pregnancies, counter to the traditional stance of avoiding it during pregnancy [131]. According to some research, the use of melatonin supplementation during pregnancy, which has been found safe in both mother and fetus according to some research, could prove to help limit complications during critical periods before and shortly after delivery [132,133]. A study suggested that preeclampsia does not have a seasonal variation, although it was observed that reduced melatonin levels were associated with the development of preeclampsia [133]. Therefore, it has been suggested that melatonin may help support a successful pregnancy.

Pregnancy is a critical time for fetal programming of hypertension. As an antioxidant therapy, melatonin may help prevent hypertension in the offspring of patients with a family history of hypertension [134]. It has been hypothesized that oxidative stress negatively impacts fertility. Since melatonin is a strong scavenger of oxidative factors, it could improve both male and female fertility and sperm and oocyte quality, resulting in increased fertilization [135,136,137,138]. Melatonin shows promise for advanced age infertility and improving IVF outcomes [139,140,141,142,143].

Delivery by cesarean may also be associated with higher levels of pro-inflammatory cytokines compared to vaginal birth [144], thereby shunting the production of pineal gland melatonin synthesis and upregulating the tryptophan conversion into the kynurenic pathway, offsetting the serotonin-melatonin pathway [145]. Women who delivered vaginally versus by cesarean had higher colostrum melatonin levels [144,146]. Finally, administering 10 mg of melatonin, compared with 5 mg or placebo, to women before the cesarean section with spinal anesthesia was shown to reduce the severity of their pain, duration of analgesic use postoperatively, and facilitated their ability to be more physically active in less time after surgery [147].

#### 3.3.2. Endometriosis

The results of supplemental melatonin in women with endometriosis are mixed. In a randomized, double-blind placebo-controlled trial, Schwertner et al. found that melatonin at 10 mg nightly reduced endometriosis pain by about 40% and reduced the use of pain-relieving medications by 80% over two months [148]. However, more recently, a small clinical trial [149] with women taking either a placebo or 10 mg melatonin during the menstrual week did not find any difference in pain reduction between the two groups. Based on the difference in findings between the two studies, a longer supplementation duration might be worth exploring in women with endometriosis.

This nutrient alone is not enough to manage the pain from endometriosis [150,151], but when it comes to pain relief, it may be a safer clinical starting point than pharmaceutical analgesics, which may have significant side effects.

#### 3.3.3. Polycystic Ovarian Syndrome (PCOS)

Polycystic Ovarian Syndrome (PCOS) is a gynecological, endocrine disorder affecting 5–10% of women. It is a multifactorial disease with increased androgens, hirsutism, acne, insulin resistance, central obesity, amenorrhea or oligomenorrhea, poor sleep, anovulation, and decreased fertility [152]. Melatonin is relevant for PCOS given that not only are there melatonin receptors on the cells as in other tissues, but melatonin is synthesized in the oocytes, ovarian follicular cells, and cytotrophoblasts of the placenta [153]. As discussed above, melatonin and its metabolites are powerful antioxidants that can preserve oocyte quality.

Both the circadian pattern and levels of melatonin are altered in PCOS. In adolescents with PCOS compared to control participants, melatonin offset is later in terms of both clock time and their wake time, while melatonin duration is longer. In adolescents with and without PCOS, later melatonin offset is associated with increased serum-free testosterone levels and worse insulin sensitivity. This finding suggests that morning circadian misalignment may be part of the pathophysiology of PCOS [154]. Other studies have found that melatonin patterns are altered in PCOS with higher serum levels but decreased follicular fluid levels, typically higher than serum levels [155]. A meta-analysis including 2553 women with PCOS and 3152 control women found that two nucleotide polymorphisms in the melatonin receptor 1A and 1B genes are significantly associated with PCOS [156].

Six months of melatonin treatment in forty normal-weight women with PCOS causes meaningful hormone changes. Androgens, free testosterone, hydroxyprogesterone, anti-Mullerian hormone, and low-density lipoprotein all significantly decreased, while there was no change in other lipid parameters or glucoinsulinemic measures. Menstrual irregularities decreased in 95% of the women [157]. This result is due to a direct effect of melatonin on the ovaries that is independent of insulin. In an eight-week trial, eighty-four participants with PCOS received either melatonin, magnesium, melatonin plus magnesium, or a placebo. Melatonin alone significantly improved subjective sleep as measured by the Pittsburgh Sleep Quality Index (PSQI) and serum high-density lipoprotein cholesterol. When melatonin and magnesium were taken together, it resulted in a significant decrease in insulin, cholesterol, low-density lipoprotein cholesterol, and testosterone levels [158]. In a randomized, double-blind, placebo-controlled study of fifty-eight women (ages 18–40 years old) they took either 10 mg of melatonin or a placebo an hour before bed for twelve weeks. Results at the end of the intervention found improvements in the melatonin group compared to placebo for mental health on the Beck Depression and Beck Anxiety Inventories. Subjective sleep quality improved on the PSQI. Lab analysis showed improvements for the melatonin group, including decreased homeostasis model of assessment-insulin resistance (HOMA-IR), serum insulin, total and LDL-cholesterol, and increased quantitative insulin sensitivity check index. Additionally, those who took melatonin had upregulation of genes for the low-density lipoprotein receptor and peroxisome proliferator-activated receptor gamma [155].

### 3.4. Gastrointestinal Health

Broad therapeutic benefits include melatonin’s role in oral care and digestive function, periodontal inflammation, post-dental surgery, and antioxidant protection against dental materials [159,160]. Studies have investigated its use in *Helicobacter pylori (H. pylori)* infections, gastric and duodenal ulcers, gastroesophageal reflux disease (GERD), and inflammatory bowel disease [161,162,163]. Melatonin and its precursor tryptophan have protective effects on mucosal tissue. A study in which *H. pylori*-infected individuals were given melatonin, placebo, or tryptophan with omeprazole is of interest. Each of the three groups had seven subjects with gastric ulcers and seven with duodenal ulcers. At the twenty-one-day mark, those treated with either tryptophan (250 mg twice daily) or melatonin (5 mg twice daily) had no ulcers, whereas the placebo group had three gastric ulcers and three duodenal ulcers. Additionally, of note is that in one study on GERD, melatonin given at 3 mg daily over eight weeks showed similar improvement in symptoms as omeprazole [163].

A study indicated that gut bacteria have a circadian clock and respond to melatonin, allowing the bacteria to synchronize with the human circadian rhythm [163]. The melatonin produced in the GI tract can, in turn, assist with gut motility and mucosal integrity via its antioxidant activity and support of the microbiome. Finally, smaller-sized studies show that melatonin can improve symptoms of pain, bloating, and constipation in individuals with Irritable Bowel Syndrome-Constipation (IBS-C) and Irritable Bowel Syndrome-Diarrhea (IBS-D) presentations. Dosing melatonin at 0.3 mg daily for IBS-C and 3.0 mg for IBS-D may benefit patients with IBS [164].

### 3.5. (Auto)Immunity

Promising, emerging research indicates that melatonin supplementation may have therapeutic benefits for autoimmune conditions, such as multiple sclerosis (MS) and perhaps Hashimoto’s thyroiditis, most likely due to its involvement in anti-inflammatory mechanisms, oxidative stress reduction, and modulation of the gut microbiota [165]. Melatonin is linked to the seasonal relapse rate in patients with MS [166]. Clinical data investigating melatonin supplementation in individuals with MS have reported better quality of life with lower doses [167] and reduced oxidative stress and inflammatory markers [168,169] in this patient population. Anderson, Rodriguez, and Reiter [170] conducted a systematic review of the correlation between the gut microbiome, gut permeability, and the possible pathophysiology of MS. An overall gut dysbiosis in patients with MS was identified as a result of increased ceramide production. The suppression of melatonin is suspected to cause this metabolic shift directly or indirectly, further complicating the circadian dysregulation that is evident in MS patients.

While not extensively studied in clinical trials, vitamin D and melatonin have been suggested to be part of a nutritional protocol for individuals with Hashimoto’s thyroiditis due to their molecular actions [171].

Currently, there is an ongoing scientific discussion regarding the use of melatonin for COVID-19. With its ability to impact mechanisms that modify immune regulation, melatonin has been included as one of the top recommendations as a preventive and therapeutic option for COVID-19, along with zinc, selenium, vitamin C, and vitamin D [172,173]. Although scientific evidence is not definitive, there are some initial indications that melatonin could be beneficial and is also considered to be safe [173,174].

#### 3.5.1. Oxidative Stress and Inflammatory States

In a systematic review and meta-analysis of thirteen clinical trials, melatonin supplementation was found to decrease inflammatory compounds (TNF-alpha, IL-6, C-reactive protein), although with a more significant lowering effect on TNF-alpha and IL-6, especially with studies ≥ twelve weeks and at a dosage ≥ 10 mg/day [175]. Athletes may be a population that experiences bouts of inflammation for which melatonin could help reduce proinflammatory mediators. In a study of oxidative stress markers in those who ran a 50 km (31 m) course, those who took melatonin had reduced levels of stress markers [176], underscoring not only the mechanism of antioxidant protection but also a practical use in athletes who are exposed to oxidative stress and inflammation that may increase their risk for vascular incidents.

Another clinical application of melatonin may be environmental toxicity. Melatonin may be one of the many antioxidants to help mitigate oxidative stress from human exposure to toxicants such as bisphenol A [177]. However, much more research is needed to understand how melatonin provides benefit relative to other antioxidants.

#### 3.5.2. Cancer Prevention and Treatment

The scientific research lineage of utilizing melatonin supplementation in those with cancer dates back at least three decades. Most notable is early research conducted on patients with solid metastatic tumors, in which it was demonstrated that high doses of melatonin were effective in arresting tumor growth and improving quality of life markers [178]. Lissoni’s group, well-recognized pioneers in the field of psycho-immune-neuroendocrinology [179], provided several reports on this dose throughout the 1990s [180] with subsequent studies confirming his findings [181,182,183,184]. Of particular mention, one of Lissoni’s studies indicated that melatonin supplementation (20 mg daily, starting seven days before chemotherapy) was helpful in chemotherapy response rate in fifty metastatic non-small cell lung cancer patients [185]. Interestingly, there was an interaction between melatonin’s efficacy and the spirituality of the patient, with greater effects noted in those with spiritual faith [185]. This intriguing finding relates to the dynamic nature of how various therapies like melatonin supplementation can be enhanced through mindset or a belief system, which is a relevant topic for immune system functioning; hence, the field of psycho-immune-neuroendocrinology.

Melatonin may help to re-establish altered circadian rhythm in cancer. Patients with breast and colorectal cancers were observed to have altered circadian rhythms associated with flattened cortisol levels throughout the day [186]. Mortality was positively associated with erratic circadian rhythm and poor sleep. Normally, cortisol levels are lowest in the evening hours and start to rise in the morning. Cortisol and melatonin work inversely, so as cortisol rises, melatonin decreases and vice versa [187]. These two endocrine messengers provide some clinical information, albeit somewhat indirectly, about the function of both the hypothalamic-pituitary-adrenal axis and the pineal gland, which is pivotal for disease outcomes like cancer where there is neuroimmune involvement [180]. Lissoni et al. also suggested that the pineal gland produces other indole hormones that could be therapeutic in cancer [188].

Under states of stress and high cortisol, tryptophan’s conversion to serotonin and melatonin is shunted to kynurenine. Often, patients with cancer experience chronic fatigue, anemia, depression, and overall decreased quality of life. Researchers recognized that individuals with solid tumors have an initial immune response involving pro-inflammatory cytokines as the body recognizes self from non-self [189]. During this process, the kynurenic pathway is accelerated, causing inflammation-mediated tryptophan catabolism, fatigue, anemia, and depression [190]. In such cases, supplementation with melatonin may be an effective way to augment the standard of care and mitigate the inflammatory cascade that ultimately leads to decreased quality of life.

In night-shift workers, circadian disruption is prevalent due to light exposure at night [191]. Upon review of human clinical trials specific to breast cancer risk in night-shift workers, researchers reported melatonin’s ability to suppress the aerobic metabolism of tumors (known as the Warburg effect) while suppressing the tumor cells’ proliferation, tumor cells’ survival, metastasis, and potential drug resistance. In human models, circadian rhythm disruption due to artificial light exposure at night significantly increased breast cancer risk [192,193]. A meta-analysis examined the role of melatonin in forty-six different microRNAs found in breast, oral, gastric, colorectal, prostate, and glioblastoma cancers. The microRNAs associated with breast, gastric and oral cancers were most responsive to melatonin treatments. Researchers identified the actions of melatonin to upregulate genes correlated to immune and apoptotic responses, where melatonin downregulated tumor cell survival involved in metastasis and angiogenesis [194].

### 3.6. Bone Health

Based on cell and preclinical data, it has been suggested that melatonin acts on both anabolic and catabolic aspects of bone metabolism [195]. Over the years, limited published clinical trials using a relatively small number of study subjects have demonstrated melatonin’s role in rebalancing bone remodeling in perimenopausal women [196] and increasing bone density in postmenopausal women with osteopenia [197]. In these studies, up to 3 mg of supplemental melatonin was used. According to findings from the year-long clinical trial referred to as Melatonin-micronutrients Osteopenia Treatment Study (MOTS), a combination of melatonin (5 mg), strontium (citrate) (450 mg), vitamin D3 (2000 IU/50 mcg) and vitamin K2 (MK7) (60 mcg) may be able to favorably impact bone markers such as bone mineral density in postmenopausal, osteopenic women, compared with placebo [195]. In addition to beneficially modifying bone markers, the intervention improved quality of life measures such as mood and sleep quality [195]. Of course, it cannot be inferred that melatonin was responsible for these effects since it was given as a combination supplement.

Investigators reviewed melatonin as a pivotal compound in age-related skeletal muscle disorders because of its involvement in mitochondrial function through its antioxidant potential [198]. Any research findings in this direction may be helpful for those with cachexia or sarcopenia. Furthermore, these authors suggested that it would be interesting to explore melatonin’s effect on the gut microbiome as it relates to skeletal muscle (the ‘gut-muscle axis’) [198].

## 4. Therapeutic Considerations

This section will address dietary melatonin and supplemental sources, including the variety of formats available, dosing, timing, contraindications, and even nutritional combinations.

### 4.1. Dietary Sources of Melatonin

Melatonin is relatively ubiquitous in nature and can be found widespread in several animal and plant foods [19,199]. Regardless of its origin and the biosynthetic pathways used to manufacture it, the chemical structure of melatonin in plants and animals is similar and bioidentical to what is found in humans [200]. In some cases, plants can be more concentrated sources of melatonin, perhaps because they can synthesize their tryptophan. Therefore, dietary tryptophan levels should be considered when assessing melatonin intake from food sources due to its biological conversion to melatonin. The important clinical point is that even though tryptophan levels in the diet may be substantially higher than that of melatonin, the conversion from tryptophan to serotonin and ultimately to melatonin may not be efficient in all individuals due to the enzymes involved. The enzyme that catalyzes *N*-acetyl-serotonin into melatonin is *N*-acetylserotonin methyltransferase (ASMT) [201]. Because methyltransferases rely on the biochemical integrity of methylation reactions in the body to transfer a single-carbon unit, the completeness of that conversion will be determined by an individual’s gene variants related to that enzyme. Methylation inefficiencies due to single nucleotide polymorphisms (SNPs) in methylation-related enzymes like 5,10-methylenetetrahydrofolate reductase (C677T) are clinically relevant in various disease states [202,203,204,205]. Unfortunately, SNPs specifically related to the ASMT enzyme have not been extensively explored in humans but would be an excellent area for translational research that could encompass clinical identification of SNPs along with nutrient modifications for therapeutic intervention [206].

Even though the amounts of melatonin in a particular serving of food may seem relatively low (on the order of an average of nanograms per gram) compared with physiological levels, there is an indication that consuming foods rich in melatonin may increase overall systemic antioxidant status. In one study with twelve healthy men [207], drinking juice extracted from one kilogram of orange or pineapple or two whole bananas resulted in significant elevations in serum melatonin and increases in antioxidant status (as measured using the FRAP and ORAC analyses). Of note, those changes in antioxidant status may be due to the melatonin content and the other vitamins and phytonutrients in the fruits. While it may not be practical to consume this quantity of juice or fruit for various reasons or due to the excessive glycemic load, it is suggestive that the diet can modify melatonin levels and, further, antioxidant status.

Research inquiries into dietary melatonin and associated health conditions have been few, perhaps due to the constraints posed by variability in melatonin content in the food supply and/or the confounding aspect of dietary tryptophan. One population-based cohort study in Japanese men (*n* = 13,355) and women (*n* = 15,724) investigated the association between dietary melatonin assessed by a food frequency questionnaire and mortality during sixteen years of follow-up between 1992–2008 [208]. Higher quartiles of dietary melatonin compared with the lowest suggested a modest effect on mortality rates.

#### 4.1.1. Plant Sources

Since its initial identification in plants in the mid-1990s, there have been subsequent references to melatonin (“phytomelatonin”) levels in various edible foods and medicinal herbs. However, its concentration is wide-ranging and inconsistent, dependent upon many factors such as cultivars, growing conditions, germination, harvesting, and processing (e.g., roasting, drying) [19,200,209]. There may also be methodological issues that result in variability in outcomes [210].

Melatonin has been documented in major plant-derived foods and beverages [211], including vegetables, fruits, nuts, seeds, grains, wine, and beers (see Table 3). Although it can be found throughout most plant parts, melatonin is typically higher in the plant’s reproductive organs, especially the seeds [200], most likely to help ensure the plant’s survival and protection against environmental stressors. Notably, one of the many roles of melatonin within plants is to stimulate the production of health-promoting phytonutrients like glucosinolates and polyphenols [212]. Along similar lines, Italian researchers have suggested that dietary phytomelatonin, rich in grains, tomatoes, grapes, and wine, may be one of the relevant phytochemicals that work in synergy with other plant-based components in the Mediterranean diet, which is currently the most well-researched healthful dietary pattern [213].

Tart cherries have been touted for their melatonin content. In one study, Montmorency cherries were found to contain 13.46 ± 1.1 ng of melatonin per gram cherries [214]. Thus, aiming for a physiological dose of 0.3 mg melatonin would imply that roughly 50 pounds of cherries would need to be consumed daily, which is an unlikely dietary goal. Even though the amount of melatonin may be nominal, tart cherries have been suggested to promote healthy sleep in insomnia, perhaps due to the minimal melatonin content, antioxidant levels, or even ability to modify tryptophan availability [215,216,217,218]. Of course, foods are complex mixtures of nutrients, so it is difficult to confirm that a physiological effect is the sole result of one compound. A recent study [219] in healthy adults found that Montmorency tart cherries as either juice (2 × 240 mL per day) or as a supplement (two capsules containing 500 mg freeze-dried tart cherry powder) for thirty days did not affect serum melatonin (although it was measured in the morning when levels tend to be low), or sleep time or quality compared with placebo. This finding may have been different in those with insomnia.

It is worth noting that there is some debate about the melatonin level of pistachios. A research paper [232] by scientists at the University of Kerman in Iran reported melatonin levels in the kernels of four different varieties of pistachio. Therein, it is claimed that one of the varieties of pistachio displayed unusually high levels of melatonin. Five years after this publication, there was a published erratum in the same journal [247] stating that the editors were informed by the German Federal Institute for Consumer Protection and Food Safety that they could not replicate these results in the same pistachios. Similarly, the American Pistachio Growers, in conjunction with researchers in the School of Nutrition and Food Sciences at Louisiana State University, reported a lower amount of melatonin in pistachios [248] when using the spectrofluorometric method used in the Oladi et al. publication [232]. While there may be some discrepancy in melatonin in plant foods due to growing and harvesting conditions, there remains some debate about whether pistachios supply this amount of melatonin. Despite these inconsistencies, pistachios still feature prominently on the list of melatonin-containing foods and may have chronobiotic potential [234].

#### 4.1.2. Animal Sources

While there are several variables to consider, in general, melatonin is found relatively less in animal foods than in plant foods. Based on published literature, milk and dairy foods, eggs, fish, and meats (beef, lamb, pork) contain some level of melatonin [211]. Conversely, animal foods tend to be better sources of dietary tryptophan than plant foods [249], so the ultimate conversion of these foods into significant quantities of melatonin may need to be considered.

### 4.2. Dietary Supplements

Melatonin has long been known to aid in sleep due to its role as a chronobiotic; however, as previously discussed in this review, there are a plethora of other benefits that supplementation may help support, including conditions with a high degree of inflammation and oxidative stress, such as in hypertension or metabolic syndrome [119,250,251]. In 2020, melatonin became one of the most sought-after dietary supplements for COVID-19 due to its role in immunomodulation and reducing the effects of the cytokine storm in conjunction with its use for sleep promotion [252,253]. Not only are more people taking melatonin as a supplement, but they are taking it in higher doses, such as greater than 5 mg, which do not have a documented long history of safe use in the general population [254].

With the recent boost in sales and widespread availability in the retail market, concerns have been expressed based on an annual report (2020) by the nation’s poison centers indicating a large number of exposure cases to melatonin was in children ≤ five years old [255]. Lelak et al. [256] investigated further into unintentional melatonin use by children and potential consequences such as hospitalization. In this report, many questions were raised, including reasons for increased exposure (e.g., more children at home during the pandemic, more accessibility of melatonin supplements in chewable or other child-friendly formats), and even proposed causes of toxicity, whether related to overdose, variability in the amount of melatonin relative to label claim, or the variety of dosing protocols. Indeed, child-proof packaging, better scrutiny of dose, and whether children should be taking melatonin supplements must be addressed. Furthermore, there is more urgent concern about the significant overage and amounts reported in melatonin supplements. The melatonin content of the 31 over-the-counter melatonin products was found to be −83% to +478% of that listed on the label. [257], with a recent lawsuit citing 165–274% of the label claim in a particular retail product [258].

The therapeutic and physiological dose of melatonin for various uses has been explored, as has the form used in supplements. As a dietary supplement in the retail channel, melatonin comes in various dosages, from as low as 0.3 mg to as high as 200 mg. As previously noted, the structure of the melatonin molecule (see Figure 6) is the same whether the source is from animals, plants, or synthetically produced.

Originally all melatonin was derived from the pineal gland of cows, sheep, or pigs. However, thirty years ago, with the development of chemically synthesized melatonin (“synthetic melatonin”), there was a dramatic shift to synthetic melatonin due to its cost-effectiveness and safety concerns over animal-derived melatonin, mainly because of the proteins or prions that could pass from cows and livestock to humans [210,259].

As previously discussed, while plants naturally contain melatonin, it is at extremely low levels, making it difficult to obtain sufficient melatonin for therapeutic doses. While the terms “plant-based” and “natural” are prevalent for marketing melatonin supplements, it is important to highlight that nearly all melatonin involves industrial processing, employing potentially toxic substrates. The distinct difference between synthetic melatonin and pure phytomelatonin in the true sense would be that the phytomelatonin supplement would exclusively involve the plant material. Melatonin originating solely from the cells of plants, without the other industrial downsides, is highly uncommon. In fact, up until 2019, there was only one phytomelatonin supplement commercially available in the U.S. with therapeutic levels of melatonin with 1 mg of melatonin per 100 mg of herbal biomass [34].

Overall, there are six aspects to consider when selecting a melatonin dietary supplement (see Table 4).

#### 4.2.1. Chemically Synthesized Melatonin

Synthetic melatonin, the most common, economical form used in dietary supplements, is produced through at least four chemical pathways using starter compounds like the following [210]:5-methoxy-3-indolylacetonitrile5-methoxy-3-(2-nitroethyl)-indole5-methoxytryptaminePhthalimide (1,3-dihydro-1,3-dioxoisonidole)

Based on the method used, synthetic melatonin can produce yields of 80–98%; however, there is the theoretical possibility that unwanted solvents and substrates could be present depending on the raw material sourced and the specifications, although the final product is often the pure substance with the other agents burned off through reactions or heat [210]. The industrial process outlined by various patents typically indicates the use of toxic solvents or petrochemical-derived substrates [263]. There are also concerns about yielding pollution and these processes’ overall negative environmental impact [201]. Because various manufacturers are aware of these constraints, different research groups are seriously attempting to streamline the steps involved. For instance, one method developed by a team in China used microwave irradiation to reduce solvent, time, cost, and pollution [263].

Thus, the purity of a melatonin dietary supplement available to most consumers on the retail store shelf may be debatable or simply unknown without greater diligence to investigate the source, which is often cumbersome and time-consuming. Moreover, with the indole structure of melatonin closely resembling that of tryptophan, there could be theoretical safety concerns related to the Eosinophilia-Myalgia Syndrome (EMS) outbreak in 1989 from the intake of at least six contaminants in an L-tryptophan dietary supplement [264,265,266], although the dose of melatonin tends to be about 1000 times lower than that of tryptophan. Common contaminants of synthetically produced melatonin, which may or may not be in the final product, include the following compounds [210]:1,2,3,4-tetrahydro-beta-carboline-3-carboxylic acid;3-(phenylamino)-alanine;1,1′-ethylidene bis-(tryptophan) (‘peak E’, one of the contaminants related to EMS);2-(3-indolylmethyl)-tryptophan;Formaldehyde-melatonin;Formaldehyde-melatonin condensation products;Hydroxymelatonin isomers;5-hydroxy-tryptamine derivatives;5-methoxy-tryptamine derivatives;N-acetyl-and diacetyl-indole derivatives;1,3-diphthalimidopropane;Hydroxy-bromo-propylphthalimide;Chloropropylphthalimde.

#### 4.2.2. Phytomelatonin

With the increased need for melatonin and safe, yet environmentally friendly formats, various manufacturers have investigated alternatives [34,267]. Although more research is needed comparing synthetic and plant-based sources of melatonin, there is some initial indication that phytomelatonin may have advantages related to improved bioavailability and efficacy. One of the unique features of phytomelatonin is that it occurs in a complex with other adjunctive plant constituents. A particular proprietary form of phytomelatonin made from alfalfa (*Medicago sativa*), chlorella powder (*Chlorella vulgaris*), and rice (*Oryza sativa*) powders have been shown to contain other phytonutrients in addition to phytomelatonin such as chlorophyll, beta-carotene, isoflavones, phytates, and saponins, all of which are naturally occurring in the plant matrix concentrate [34] (see Figure 7).

One of the interesting benefits of phytomelatonin over an isolated, synthesized chemical compound of melatonin is the complex, diverse composition of phytochemicals in the plant matrix concentrate. For example, it contains small amounts of the xanthophyll carotenoids, lutein, and zeaxanthin, which are known to concentrate in the back of the eye, specifically in the macula and fovea [268]. Research suggests that these plant compounds may help protect the eyes by absorbing harmful blue light [268,269]. Therefore, not only does phytomelatonin supply the bio-identical melatonin to help with circadian rhythm imbalance at night, but it also adds to the photoprotective compounds for the eye to shield against blue light, making for a complete multi-functional approach.

In fact, those additional constituents may completely or partially explain why this phytomelatonin outperformed synthetic melatonin in cellular assays for inflammation (using COX-2 inhibition) and free radical scavenging [34] (see Table 5).

**Table 5 nutrients-14-03934-t005:** Comparison between phytomelatonin and synthetic melatonin. * The phytomelatonin used for comparison is the proprietary format utilized in [34].

Feature	Phytomelatonin *	Synthetic Melatonin
Origin	Plants	Chemicals
Processing	Customized cultivation technique of selecting the ideal location, soil, climate and optimal method/time to harvest based on the plant’s cycles to optimize melatonin levels	Chemical synthesis
Constituents	Bioidentical melatonin plus other plant actives; no excipients, fillers, or binding agents	Bioidentical melatonin and possibly contaminants from the chemical synthesis; depending on the dietary supplement, it may contain excipients, fillers, or binding agents
Environmentally safe?	Yes	No, uses toxic solvents and generates pollution
Other *nutritionally active* compounds included	(Essential) Fatty acids, amino acids, vitamins (vitamin K, riboflavin (vitamin B2), choline, vitamin E, thiamin (vitamin B1), pyridoxine (vitamin B6), biotin), minerals (trace amounts of calcium, magnesium, zinc, iron, manganese, selenium, copper, potassium, sodium, phosphorus, chloride, iodine), phytonutrients (beta-carotene, xanthophyll, zeaxanthin, lutein, chlorophyll, violaxanthin); Concentration of these adjunctive compounds depend on growing and seasonal changes.	None
Anti-inflammatory activity	Yes, more effective in inhibiting COX-2 in a cellular assay compared with synthetic melatonin [34]	Yes, although not more effective than phytomelatonin * [34]
Antiradical scavenging activity	Yes, it possesses significantly stronger free radical scavenging capacity as compared to synthetic melatonin using a cellular assay to assess Free Radical Scavenging Percentage (DPPH%) [34].	Yes, it has antiradical scavenging activity, although less than phytomelatonin * [34].
Oxygen Radical Absorbance Capacity (ORAC) (see Figure 8)	17,200–18,500 [270]	1932, 4492 [271] 4830 [272]

### 4.3. Dosing

A generally accepted guideline is to use the lowest effective dose of melatonin as the most appropriate course [273]. Larger doses do not always confer greater health benefits. The upper limits of a lethal dose of melatonin have not been clinically established. A recent systematic review and meta-analysis of safety studies on high-dose melatonin (≥10 mg/day) in adults concluded that there are limited studies from which to draw any solid conclusion about its safety profile [274].

In clinical applications, too much melatonin or various extended-release formats have been documented to produce side effects such as amnesia or a “melatonin hangover” the next day, finding it harder to fall asleep, or sleeping well for three to four hours and then waking up and not being able to go back to sleep [275]. Those with certain genotypes, such as polymorphisms in the melatonin receptor 1B (*MTNR1B)* gene, may require monitoring of hemoglobin A1C if they take supplemental melatonin [275,276].

Within clinical medicine, it has also been more anecdotally discussed whether high doses over the long-term can negatively impact the body’s melatonin production, with individuals potentially becoming dependent over time. While a theoretical concern, there is a lack of significant scientific evidence that this dependence can develop, and if it does, under which conditions and individuals may be most susceptible. There have also been reports of vivid dreams or nightmares while using melatonin supplements [164]. Since the average human adult produces between 0.1 mg and 0.9 mg of melatonin daily [277], this range is known as physiological doses. Amounts above this range are known as pharmacological doses. Much has been written about melatonin’s therapeutic value, but the doses used in studies tended to be supraphysiological or based on previous studies that did not have an explicit rationale for choosing the amount. Therefore, some dogma about dosing melatonin has developed in scientific research and clinical medicine.

In a pivotal study by Zhdanova et al. [278], multiple doses were compared: a physiological dose (0.3 mg), a pharmacological dose (3 mg), and a low physiological dose of 0.1 mg. They found the best objective data at 0.3 mg of melatonin. Sleep data were obtained by polysomnography. The physiological dose (0.3 mg) restored sleep efficiency and elevated plasma melatonin levels to normal during early adulthood. The pharmacological dose (3 mg), like the lowest dose (0.1 mg), also improved sleep; however, it induced hypothermia and caused plasma melatonin to remain elevated into the daylight hours. Interestingly, the control group (not insomniacs) also had low melatonin levels, but melatonin did not improve sleep. The low dose in the study did not raise melatonin levels into the normal range. It is an intriguing point that we need to lower our body temperature to sleep well but doing so excessively can disrupt sleep. Melatonin’s action of lowering body temperature is important to monitor and may give significant clues to the appropriate dosage. Symptoms like needing more blankets, or excessive movement, may indirectly suggest an imbalance of melatonin.

The dose of melatonin was patented up to 1 mg based on this earlier research [279], excluding dietary supplement manufacturers from selling this dose and thus, using higher doses. It can be postulated, in part, that the earlier historical use of higher-dose melatonin levels may have been associated with the inability to operate freely with known efficacious doses of melatonin in supplemental format.

Lissoni et al.’s cancer research from over twenty years ago demonstrated that 20 mg given intramuscularly followed by oral daily doses of 10 mg was effective in arresting tumor growth and improving quality of life markers [178]. Higher oral doses (50 mg daily) were given in a subsequent study with immunotherapy to 14 patients with advanced hepatocellular carcinoma [280]. Other studies since these pivotal publications have used this dose as a reference [181,182,183,184]. Little research has been conducted on lower doses to determine if they are as effective in cancer patients or if the physiological dose of 0.3 mg can be used for prevention. Hopefully, future studies will delve into these questions.

In 2002, Lewy et al. found that physiological doses (0.5 mg) may offer benefits that pharmacologic doses (20 mg) do not [281,282]. They observed the effects of the dosage of melatonin in blind humans who often have disrupted circadian rhythms due to the pineal gland not receiving appropriate stimulation from the retina. They concluded that that too much melatonin may negatively affect the melatonin phase-response curve [281]. The phase-response curve describes how an intervention given at various times in relation to the individual’s sleep period will influence their circadian rhythm, i.e., whether the intervention will cause a phase delay, advance, or no phase shift. This point supports the concept that too much melatonin may not benefit a person. It also begs the question, “How much is too much?” It will be difficult to answer this question for the masses as hormone production, genetics, and timing of secretion are complex variables to assess for the individual. In addition, the bioavailability of the form or product used can play a part.

Anecdotally, clinical use of supplemental melatonin has commonly ranged between 1–3 mg as a daily dose; however, some find that the dose is determined by the product type, with lower doses being just as effective in some instances. Melatonin is quickly broken down by the body and should be dosed at the appropriate, personalized level for each patient. Controlled-release formulations have been discouraged in older adults due to the possibility of prolonged melatonin levels and unknown implications thereof [273]. Because it is rapidly metabolized, melatonin might be used more regularly daily by those who need it, rather than every second or third day. In other words, if a person has symptom relief from 1 mg of melatonin, they may not experience the same benefit with 3 mg of melatonin every third day. Products containing a dose of 3–5 mg are often chosen because they are perceived as a good value but supplementing with more melatonin than physiologically required is not always better if the dose is incorrect. For this reason, starting at the physiological dose of 0.3 mg and increasing if necessary is often recommended except for specific conditions where higher doses are therapeutically prescribed, such as jet lag, shift work, or cancer.

For jet lag, recent studies are lacking in detailing the most efficacious dose. Most research on the use of melatonin for jet lag is from 10–20 years ago or longer. Melatonin may not be effective for everyone but could be well-suited for those individuals with a history of jet lag and/or for those flying across ≥ five time zones to the East [74]. Upon arriving in a new time zone, it is advised to follow the new time zone sleep cycle, taking the oral melatonin supplement thirty to sixty minutes before desired sleep on the first night and continuing for the following three to four nights, gradually reducing the dose. In one study, the fast-release formula was more effective than the slow-release preparation [283]. Some studies used a protocol in which melatonin supplementation was initiated three days before travel started [284,285]. Supplemental melatonin is intended for short-term use: 3–6 days after arrival at destination.

Similarly, study findings are inconsistent for shift work. Older clinical studies with smaller subject numbers found no difference [286,287] or modest differences [288] in sleep outcomes compared with placebo. A meta-analysis [289] found essentially no benefit from melatonin.

The use of melatonin in children is now widely accepted for various disorders, but since the studies range so widely in dosing, a critical analysis is required for each child. Dyssomnia, ADHD, and ASD have been studied and reviewed, all confirming effectiveness and safety of melatonin. It should be noted that studies were of various lengths, with some as short as two weeks, and the longest-lasting study was six months. Only one questionnaire-based study investigated long-term melatonin use in children with ADHD and chronic insomnia and, by its design, was based on subjective symptom reports. In an average of 3.7 years of follow-up from previous clinical trial participation of pharmacological doses of melatonin, 65% of children were still using melatonin as prescribed in the study, but only 9% were able to discontinue use [290]. Neither the parents nor children monitored their melatonin use but relied on the initial study proposal, so dosing varied widely. Compliance nearly four years later was at 65%, presumably due to the satisfaction of use.

Dosing parameters may vary according to factors such as the child’s medical problems, severity, type of sleep problems, or the associated neurological pathology. Indiscriminate, long-term dosing may lead to unnecessary dependence and even perturbations in puberty onset when nocturnal levels of melatonin tend to decline [291]. More discriminating research is warranted to understand the implications of exogenous melatonin in children, pre-teens, and teens. Based on what is currently known and the easily administered formats provided in the market (e.g., gummies, chewables), it would seem prudent to exercise caution. The authors would like to emphasize that the dramatic increase in the use of supplemental melatonin in children is unwarranted because they endogenously produce more melatonin than adults [10]. Furthermore, it seems that the research has mistakenly been construed and applied to healthy children rather than children with particular needs, such as autism or ADHD.

Finally, to underscore the discussion of dosing in both pediatrics and adults, it is important to understand that melatonin is metabolized via the liver primarily by the enzyme CYP1A2 [292]. The slow metabolism of this enzyme has clinical applications. A melatonin clearance test is reasonable but difficult to implement practically. Therefore, loss of response after several weeks may suggest a patient’s tolerance of melatonin and necessary dose reduction. In one report, clinicians observed that the initial response to melatonin had disappeared weeks after starting treatment, and that the favorable response returned with dose reduction [293].

In conclusion, what is optimal dosing of melatonin with the current understanding? Based on the review of current literature, it would seem that the administration of melatonin, as with other exogenous hormones like estrogen [294], would be at the lowest dose for the shortest period of time. Moreover, with melatonin specifically, the physiological kinetics and circadian rhythm need consideration, ideally taking a physiological dose (0.3–0.5 mg) of melatonin with sustained release properties and going into darkness a minimum 30 min before desired sleep, thereby mimicking the physiological kinetics based on light and dark patterns as well as the typical endogenous trajectory of immediate and then sustained release over four to five hours (See Figure 9). The caveat to this dosing recommendation is in relation to the use of melatonin for acute treatment for cancer or other health conditions that may require higher dosing or other forms of delivery (e.g., intramuscular, intravenous, etc.).

### 4.4. Timing

There are some general opinions on dosing melatonin from a timing perspective. Historically, much of the research related to sleep has indicated melatonin supplementation thirty to sixty minutes before bed; yet, some studies have shown that it can be taken up to four hours prior and be effective. Lewy suggests that dosing for sleep requires a minimum of twelve hours of wakefulness [281]. It has been recommended to take melatonin in conjunction with being in darkness versus light stimulation from television and computers [281]. In those instances, it is often more practical to adhere to the thirty to sixty minutes before bed.

### 4.5. Bioavailability

Theoretically, much like other nutrients, several factors could affect the bioavailability of a melatonin supplement, including overall reduced digestion and absorption capacity, the composition of the gut microbiome, genetic variability in enzymes that metabolize melatonin, and/or the excipient profile of the dietary supplement, to name a few. It is well accepted that melatonin in tablet or capsule form is generally not very bioavailable (1 to 74%) and can be influenced by multiple variables [295]. A systematic review by researchers at the University of Copenhagen cited an average of 15% bioavailability from oral melatonin, which was highly variable (range: 9–33%) and affected by age, illness, caffeine, smoking, and medications [296].

From a pharmacokinetic standpoint, Vasey, McBride, and Penta [297] indicated that melatonin administered in tablet or capsule forms follows first-order or concentration-dependent kinetics. This implies that the kinetic absorption of melatonin is, in fact, not saturable, meaning that larger doses would produce higher plasma concentrations. Oral melatonin has been studied to reach peak plasma concentration after forty-one minutes of administration, although it was reported that melatonin pharmacokinetics varied greatly between individuals [297].

Because of its well-recognized low bioavailability, scientific innovations are underway to develop formulations that bypass the gastrointestinal tract, have a longer half-life, and/or leverage nanotechnology [261,298,299]. Overall, more research is needed to better understand the bioavailability of various formats and sources of melatonin, as well as innovative developments to optimize its physiological effects [200]. Another area of research to investigate is how the different sources of supplemental melatonin may vary in uptake. For example, whether a phytomelatonin form in its natural plant complex is more readily absorbed than one chemically synthesized. At this junction, these questions remain unanswered.

### 4.6. Contraindications and Combinations

#### 4.6.1. Contraindications

Oral melatonin is mostly perceived as safe based on published reports, even in higher doses [300]; however, there can be specific instances that may necessitate further diligence and clinical oversight. Earlier research has identified that the majority of melatonin is metabolized through cytochrome (CYP) 1A2 [292,301], with lesser metabolic activity through CYP1A1, CYP1B1, and CYP2C19 [292,302]. Therefore, medications that affect these enzymatic pathways will influence melatonin metabolism. Caution should be exercised when melatonin is taken with one or more medications; otherwise, adverse side effects (like extreme sedation) can result [303]. For example, one of the most well-known interactions is melatonin with the anti-depressant medication, fluvoxamine. This pharmaceutical is a known inhibitor of CYP1A2 and can result in the potentiation of melatonin levels due to reducing its degradation [304]. Similarly, caffeine is also metabolized through CYP1A2 and can increase melatonin levels [305].

In summary, while not a comprehensive listing, melatonin can interact or be influenced by pharmaceuticals, nutrients, or herbs with blood thinning, blood sugar-lowering, blood pressure reducing, anti-convulsing, sedative, anti-depressant, and/or immunosuppressive activities [306]. Importantly, melatonin metabolism and activity can be potentially impacted by many dietary, supplemental, and pharmaceutical agents, all of which add to the fact that there is variability in an individual’s personalized response to this compound based on their dietary and medical context. Historically, due to the lack of safety data, melatonin supplementation was not recommended for women who are pregnant or nursing. However, there is some indication that melatonin may be of benefit during both these stages. Therefore, having these women consult with a health professional who understands the patient’s personalized needs is key. Similarly, people diagnosed with any disease should consult their health professionals and take it under medical supervision [306].

While there are many reasons to be concerned about interactions, there may also be a role for melatonin to offset the toxicity of certain drugs through its antioxidant activity [307,308]. Thus, more research and clinical oversight into using exogenous melatonin are warranted. Ongoing information on melatonin and related topics can be found in Table 6.

#### 4.6.2. Combinations

As stated above, combining melatonin supplementation with actives that influence melatonin pharmacokinetics may cause side effects. Conversely, in certain instances, nutrients may be more effective when included with melatonin or vice versa. In this section, combinations of melatonin with (1) vitamin C, (2) vitamin B12, and even (3) myo-inositol, folic acid, and vitamin D3 will be detailed. Currently, there is a lack of significant data to make therapeutic suggestions about combinations.

##### Vitamin C

Because melatonin has antioxidant capacity, its actions may be complementary to other types of antioxidants. In a cell assay, vitamin C and phytomelatonin have been shown to demonstrate significant increases in free radical scavenging activity compared to either by themselves [34]; however, those results need to be tested in a clinical trial to be able to extrapolate those findings to humans.

##### Vitamin B12

A few human clinical studies published more than two decades ago would suggest that vitamin B12 supplementation increased the phase advance effect of morning light and decreased nocturnal melatonin levels [309,310], and in the case of methylcobalamin specifically, may even result in less need for sleep and impart better sleep quality [311]; however, the wider implications for this effect remain largely unexplored.

##### Myo-Inositol, Folic Acid Vitamin D

Promising clinical data indicates that melatonin may help enhance oocytes’ quality. One study found that adding melatonin to a supplemental protocol of myo-inositol and folic acid may help increase oocyte viability in women undergoing IVF [312]. Combining myo-inositol and folic acid with added melatonin increased oocyte quality in women with PCOS compared with only using myo-inositol and folic acid or folic acid by itself [313]. Finally, when added to a regimen with vitamin D3, melatonin and myo-inositol were beneficial in oocyte quality, fertilization, and pregnancy outcomes in women undergoing an intracytoplasmic sperm injection [314].

##### Glutathione

Glutathione is one of the major antioxidant systems in the body and is susceptible to changes in redox status and nutrient supply [315]. Various research studies have confirmed that there are individual associations between melatonin and glutathione [165], and glutathione and vitamin D [316]. Thus, while not yet extensively researched, there could be an interrelationship between the collective antioxidant activities of glutathione, vitamin D, and melatonin. Parsanathan and Jain [316] have demonstrated the effects of glutathione deficiency in high-fat-fed mice with observed epigenetic changes in genes responsible for vitamin D metabolism. Type 2 diabetics may be particularly vulnerable to decreases in antioxidants and increases in inflammatory markers. One recent clinical trial in type 2 diabetics found that modest vitamin D supplementation helped to increase both vitamin D and glutathione levels, while decreasing oxidative stress and inflammation [317]. More research is required to understand how glutathione levels are implicated with both melatonin and vitamin D.

### 4.7. Lifestyle Aspects

#### Blue-Light-Blocking Glasses

The circadian release of melatonin is strongly suppressed by light, particularly blue light. Even light of five to ten lux while sleeping with eyes closed will impact the circadian system [318]. Blue-light-blocking glasses can protect from the melatonin suppressing effects of light [319]. When worn during a 60 min light pulse at 0100 h, blue-light-blocking glasses resulted in a slight increase in melatonin levels compared to baseline, while melatonin levels decreased significantly by 46% in the control condition [320]. After seven nights of wearing blue blockers for two hours before bed, people with insomnia have significantly improved subjective and objective total sleep time, sleep quality, and soundness [321]. For people with Delayed Sleep-Wake Phase, wearing blue-light-blocking glasses from 2100 h to bedtime for two weeks resulted in their dim-light melatonin onset occurring 78 min earlier, while sleep onset was significantly earlier by 132 min [322]. In healthy adults without sleep or circadian disorders, using blue-light-blocking glasses from 1800 h until bedtime for a week resulted in subjective reports of earlier sleep onset, though objective measures were not improved.

### 4.8. A Comprehensive Clinical Approach to Melatonin

For those with allopathic, functional medicine, holistic, integrative, naturopathic, and/or nutritional clinical orientation to assessing and treating patients, there are many approaches to addressing endocrine or, more specifically, melatonin-related imbalances or dysfunction. It would be worthwhile to investigate the constitutional aspects such as genetic components that would result in modified responses in receptivity or production to or of melatonin, or even acute or chronic triggers that could necessitate alterations in requirements for melatonin, such as increased inflammation, toxic exposures, stressful events, and others. Concerning body systems, all may be impacted mechanistically and, ultimately, give rise to corresponding clinical conditions. Finally, perhaps most importantly is an evaluation of lifestyle factors such as sleep habits and patterns, physical activity influences on oxidative stress or inflammation, the role of melatonin in the diet and through supplementation, impacts of stress on the psychoneuroendocrine system, and modifying one’s response through modalities such as meditation, breathwork, and creative arts, and, a consideration of one’s relationships and cultivating a healthy community for health and well-being. A summary of these elements can be found in Table 7.

#### Laboratory Testing

From a clinical perspective, assessing one’s levels of systemic melatonin is difficult. Salivary melatonin samples are collected in the evening to determine the time of melatonin onset, called “dim light melatonin onset” (DLMO). Urinary melatonin metabolite 6-sulfatoxymelatonin can be measured in morning urine to assess accumulated melatonin over the night. Blood samples can also be collected, showing melatonin levels in real-time, and is a more sensitive measure, so preferred in cases of low melatonin.

Routine laboratories do not tend to offer this type of testing, and of the specialty labs that offer these assessments, there are some confounding factors to consider:

(1) *Light exposure*: Results will be variable and potentially influenced by light conditions. Since melatonin is strongly suppressed by light, the test will not yield many useful data in people’s normal home lighting. It will simply reflect the suppressed levels expected in electric lighting. For testing, it is recommended that light levels be 30 lux for four to seven hours before bed and red light exposure if possible. The importance of being in dim light is not often communicated in patient instructions or is widely known by health practitioners;

(2) *Timing*: To get information on the patient’s melatonin onset time, they need to collect samples every thirty or sixty minutes for four to seven hours while in dim light;

(3) *Diet:* There is still insufficient data about how dietary (exogenous) and gut/pineal (endogenous) sources of melatonin interact and how measurements could be implicated by potential cross-talk among the melatonin inputs. To reduce the risk of inaccuracies, the patient would at least need to abstain from dietary sources of melatonin; however, this is not always practical, and many foods contain melatonin. Furthermore, the timing of eating and time of sleep may influence melatonin levels, as can intermittent fasting. It remains unclear how food intake (types of foods, macronutrient distribution, timing) during the day can alter pineal gland secretion of melatonin at night;

(4) *Other factors:* Season, physical activity, age, gender, genetic factors related to the volume of the active pineal gland, and possibly posture at the time of sample collection may all influence melatonin levels [323].

In summary, there are many variables to consider with a laboratory assessment that may lead to inaccuracies. Moreover, since people are not likely to adopt the lifestyle of being in such low evening light, the utility of such a test seems limited unless the practitioner wants to verify that the patient’s pineal gland can make melatonin when they are in dim light conditions.

## 5. Conclusions

Overall, melatonin is an intriguing compound, not unlike vitamin D, which is pleiotropic in activity and responsive to light-dark cycles. From a scientific perspective, melatonin acts as a powerful antioxidant that can cross the blood–brain barrier, inhibit inflammation, and interact with the gut microbiome. From a clinical point of view, melatonin imbalance may indicate “darkness deficiency” in much the same way that vitamin D may infer whether or not someone has a “light deficiency”.

To some extent, melatonin has been misunderstood as a sleep aid. It has been shown to have systemic effects through its mechanisms of action, ultimately having the potential for a significant impact on the etiologies of multiple chronic diseases. More clinical research is warranted to better understand melatonin’s effects from dietary intake and how it is influenced by gene variants and receptors, as well as its metabolism, metabolites, and laboratory assessment. When it comes to supplemental sources, phytomelatonin may have greater efficacy and present a more sustainable, low-toxic option compared with synthetic formats of melatonin. Finally, lifestyle habits such as proper exposure to light and darkness and less blue light at night are also helpful for establishing healthy melatonin levels. Based on all these factors, clinicians can put together nutrition and lifestyle recommendations to optimize melatonin for their patients (see Figure 10).

In the future, from a clinical perspective, it will remain increasingly important to identify how melatonin addresses the physiological needs of an individual, taking into account diet, medications, dietary supplements, and lifestyle.

## Figures and Tables

**Figure 1 nutrients-14-03934-f001:**
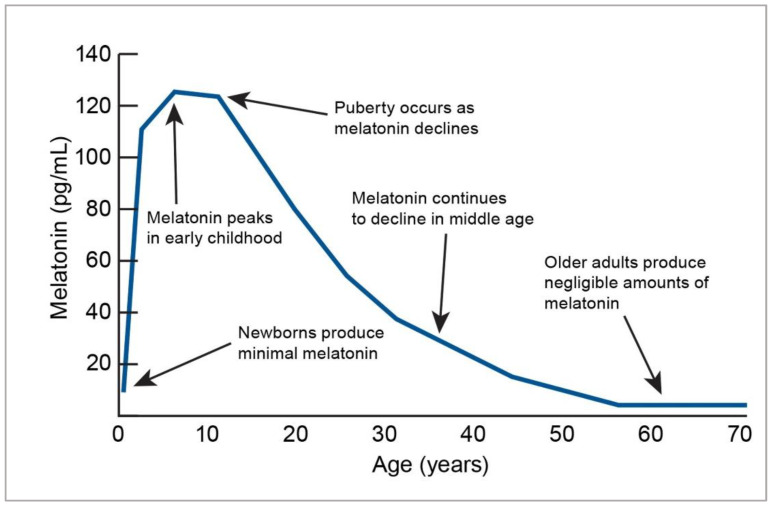
Age-related decrease in melatonin in humans. Modified from [10].

**Figure 2 nutrients-14-03934-f002:**
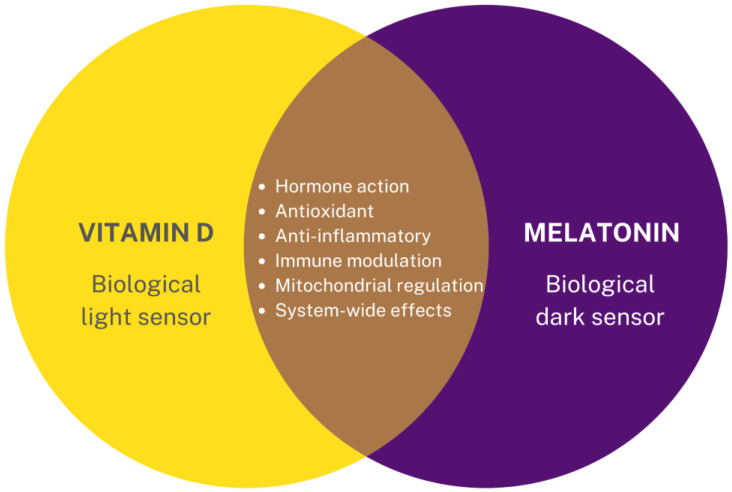
Vitamin D and melatonin as light and dark sensors with shared functions. Graphic created using https://Canva.com accessed 27 July 2022.

**Figure 3 nutrients-14-03934-f003:**
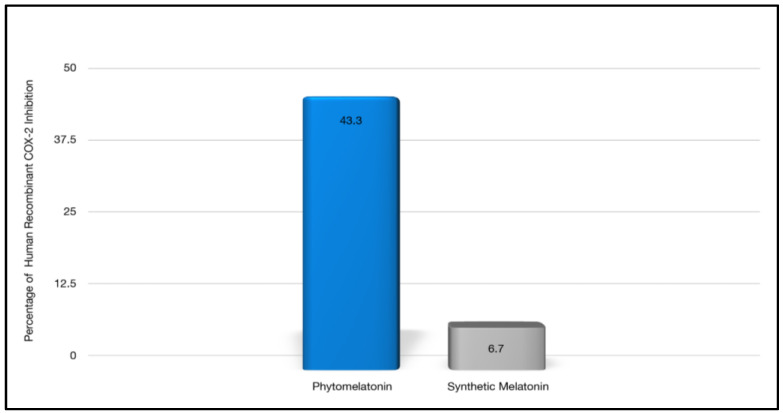
Inhibition of inflammation by phytomelatonin (blue bar) and synthetic melatonin (gray bar). Data are expressed as a percentage of human recombinant COX-2 inhibition. Amounts used for each were 0.030 mL (5 mg/mL). Values are derived from the original data presented in [34].

**Figure 4 nutrients-14-03934-f004:**
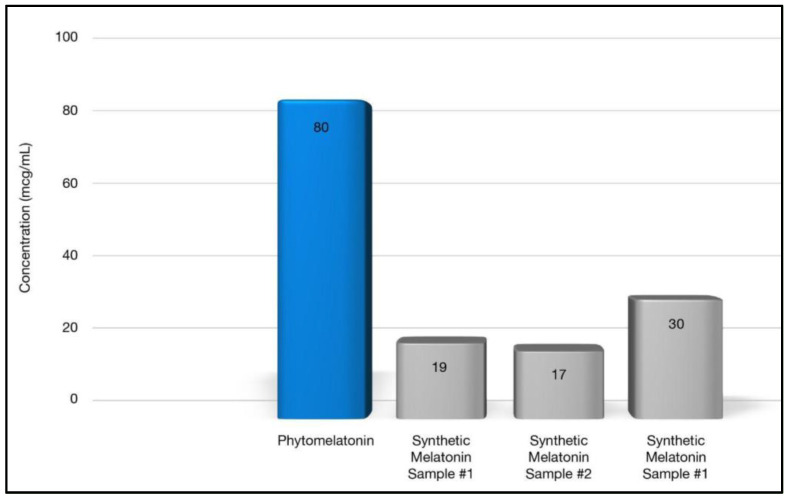
Free Radical Scavenging Percentage (DPPH%) by phytomelatonin (blue bar) and three synthetic melatonins (gray bars). Data are expressed as mcg/mL. Values are derived from the original data presented in [34].

**Figure 5 nutrients-14-03934-f005:**
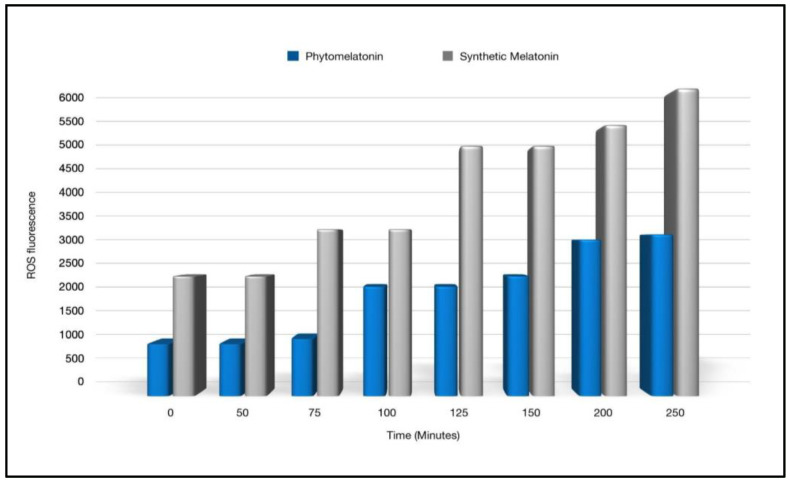
ROS fluorescence in human skin cell line by phytomelatonin (blue bars) and synthetic melatonin (gray bars). Data are expressed as ROS fluorescence using 50 mcg/mL for both phytomelatonin and synthetic melatonin. Values are derived from the original data presented in [34].

**Figure 6 nutrients-14-03934-f006:**
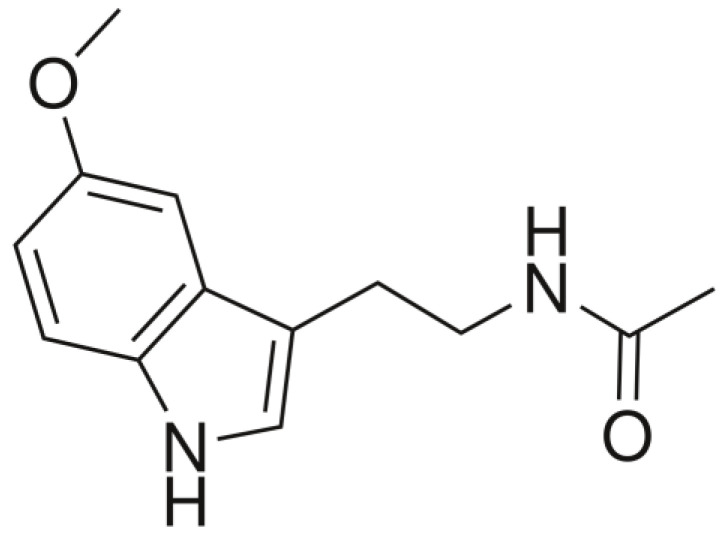
Chemical Structure of Melatonin (CAS: 73-31-4; Wikipedia, public domain) [254].

**Figure 7 nutrients-14-03934-f007:**
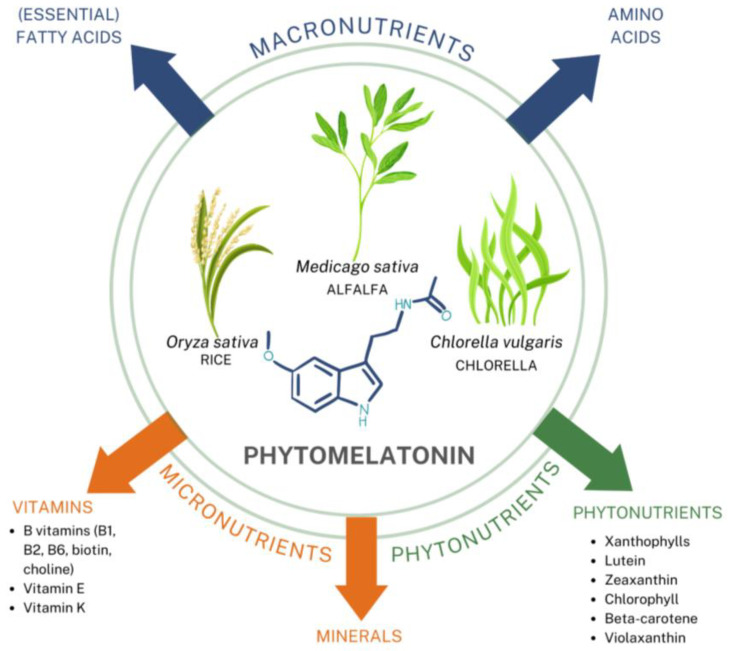
Nutritional characteristics of phytomelatonin. The phytomelatonin detailed below refers to the proprietary format utilized in [34]. Graphic created using Canva.com using images from pavelnaumov (chlorella, alfalfa), Victoria Sergeeva (rice), and Walrus_d’s (melatonin). https://Canva.com accessed 27 July 2022.

**Figure 8 nutrients-14-03934-f008:**
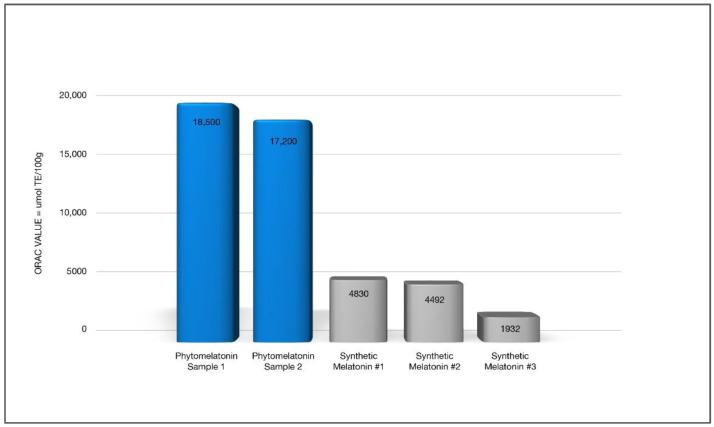
Oxygen radical absorbance capacity (ORAC) of two samples of phytomelatonin (blue bars) and three types of synthetic melatonin (gray bars) [271,272].

**Figure 9 nutrients-14-03934-f009:**
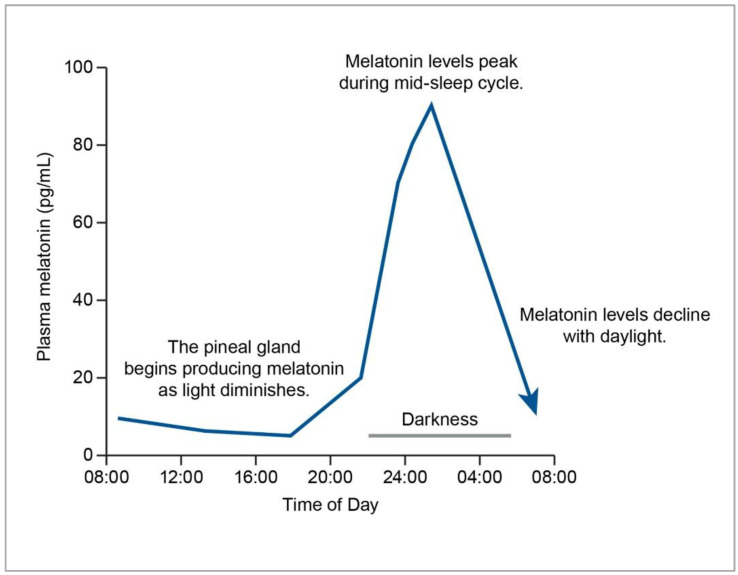
Melatonin production throughout the day. Modified from [10].

**Figure 10 nutrients-14-03934-f010:**
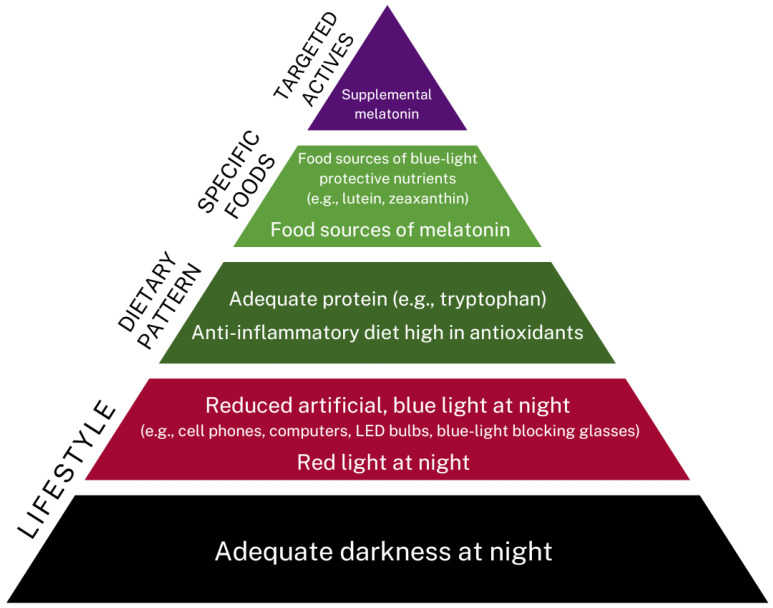
A comprehensive nutrition and lifestyle approach to optimizing melatonin. There are several aspects to ensuring healthy melatonin levels, including lifestyle modifications involving light exposure, selecting specific dietary patterns and foods, and, when required, targeted supplemental sources. Graphic created using https://Canva.com, accessed 27 July 2022.

**Table 1 nutrients-14-03934-t001:** Comparison of vitamin D and melatonin.

Feature	Vitamin D	Melatonin
Basic functions	Considered to act as a hormone; Antioxidant; Anti-inflammatory compound; Mitochondrial regulator	Hormone; Antioxidant; Anti-inflammatory compound; Mitochondrial regulator
Bodily systems	All	All
Relationship with light	Light (UV) is needed for synthesis.	Darkness is needed for synthesis.
Synthesis	Synthesized in the skin, activated by liver and kidney	Synthesized in the skin and many other tissues; Produced by pineal gland and gut (enterochromaffin cells)
Seasonal variation	Yes	Yes [28]
Chemical nature	Lipid-soluble	Amphiphilic
Transport	Crosses blood–brain barrier	Crosses blood–brain barrier
Nutritional status	Greater risk of insufficiency and/or deficiency with increasing age	Greater risk of insufficiency and/or deficiency with increasing age
Obtained from dietary sources	Yes	Yes
Biological need may change depending on lifestyle	Yes	Yes

**Table 2 nutrients-14-03934-t002:** Summary of Possible, Personalized (Select) Clinical Uses for Melatonin. Note that this list is not exhaustive; and that there are varying levels of evidence for each condition.

Body System	Possible Clinical Uses
Central Nervous System	Circadian rhythm modulation Sleep-wake disorders Sleep disturbance Cognitive conditions such as dementia Migraines and headache Tinnitus Attention-Deficit Hyperactivity Disorder (ADHD) Autism Eye disorders (e.g., glaucoma)
Cardiovascular System	Hypercholesterolemia Hypertension/high systolic blood pressure Metabolic syndrome Endothelial dysfunction Glycemic balance (varying effects due to differing response in MTNR1B G-risk allele carriers)
Reproductive System	Preeclampsia Fertility As an adjunct to care for endometriosis Polycystic Ovarian Syndrome (PCOS)
Gastrointestinal System	Gastroesophageal Reflux Disease (GERD) Ulcers Irritable Bowel Syndrome (IBS)
Immune System	Autoimmune conditions (Multiple sclerosis, Hashimoto’s thyroiditis) Coronavirus Disease (COVID-19) Oxidative stress from athletic performance stress Oxidative stress from excessive environmental toxin load Cancer; chemopreventive and as an adjunct to treatment depending on the cancer type and individual
Musculoskeletal System	Osteopenia

**Table 3 nutrients-14-03934-t003:** Select plant food sources of melatonin.

Category	Select Types (Listed in Alphabetical Order)	References
Vegetables	Several types: Asparagus, beetroot, cabbage, carrot, corn, ginger root, purslane, spinach, taro	[209,211,220,221,222,223,224,225]
Fruits	Several types: Apple, banana, cherries (sweet, tart), cucumber, grapes, kiwifruit, peppers, pineapple, pomegranate, strawberries, tomatoes	[220,221,222,225,226,227,228,229,230,231]
Nuts	Almonds, pistachios, walnuts	[224,225,226,232,233,234]
Seeds	Anise, celery, coriander, fennel, fenugreek, flax, green cardamom, mustard (black, white), poppy, sunflower; Raw and germinated seeds of alfalfa, broccoli, lentil, mung bean, onion, red cabbage, and radish	[209,225,235,236]
Grains	Barley, oat, rice, wheat	[221,222,224,225]
Beans Legumes	Kidney beans (sprouts), soybeans	[237,238]
Herbs Spices	Black pepper, feverfew, sage, St. John’s wort, select Chinese medicinal herbs	[239,240,241,242]
Oils	Argan oil, extra virgin olive oil, grapeseed oil, linseed oil, primrose oil, sesame oil, soybean oil, sunflower oil, walnut oil, wheat germ oil	[243]
Beverages	Beer, coffee, grape juice, orange juice, wine	[226,231,244,245,246]

**Table 4 nutrients-14-03934-t004:** Considerations in the selection of a melatonin supplement.

Factor	Details	General Comments
Source	Animal (pineal gland) Chemical synthesis Phytomelatonin Microbial fermentation products (bioengineered)	Synthetic melatonin is the most common form of melatonin on the market but can result in the use of potentially unwanted solvents and substrates in addition to it being environmentally undesirable [210]. Plant-based melatonin presents challenges in concentrating to a viable dose of melatonin. Animal-based melatonin can involve the risk of viral infections. Microbial fermentation products are under development.
Route	Oral intake Oral, immediate release Oral, sustained, time-release Sublingual Intravenous Intramuscular Intranasal Transdermal Anal/Suppository Vaginal delivery	There are a variety of formats available, and each needs to be individualized to the person’s needs. Several newer formats are being developed for optimizing delivery, although only oral administration is considered a dietary supplement in the U.S. [260,261].
Delivery	Capsule Tablet Chewable Gummies	A trending format is that of gummies, which is a sweetened gelatinous-type delivery for greater palatability. While it may be the desired delivery form for consumers, there are concerns about the stability of melatonin in such a hygroscopic matrix, the resulting sugar content, the addition of dyes or flavoring agents, and the potential for an overdose of melatonin, especially in the case of children.
Actives	As an isolated compound In combination with other actives In a plant matrix with other phytonutrients	Often, dietary supplements of melatonin will include other nutritional or herbal actives with the intention of synergy or improved efficacy, although, on the whole, these types of preparations have not been effectively studied for interactions.
Quality	Certified Good Manufacturing Practices (cGMP) Third-party testing for heavy metals, and contaminants Packaging integrity to ensure shelf-life and stability.	Not all dietary supplements have the same quality. cGMP and third-party testing can be markers of objective quality measures. Melatonin can degrade in the presence of air and light, so minimizing exposure [262] in oxygen-barrier blister packs would be preferential over open bottle format.
Dose	Physiological dose (0.3–1.0 mg) Supraphysiological dose for occasional use (≥3 mg) Therapeutic dose prescribed by a qualified healthcare practitioner	There is much debate about proper dose levels. Consider safety in addition to efficacy for the clinical condition it is being used for in a patient, as well as the duration of use, whether low dose, short term or high dose, long term.

**Table 6 nutrients-14-03934-t006:** Select Clinical Resources and Reference Sites for Information on Melatonin (Accessed on 27 July 2022).

Category	Website
Clinical dosing recommendation and contraindications	Natural Medicines Comprehensive Database: https://naturalmedicines.therapeuticresearch.com/ accessed on 27 July 2022
Research sites	Melatonin database of studies including phytomelatonin: https://www.phytomelatonin.com accessed on 27 July 2022 Melatonin Research: https://www.melatonin-research.net/index.php/MR accessed on 27 July 2022 National Center of Complementary and Integrative Health: https://www.nccih.nih.gov/ accessed on 27 July 2022
Professional Organizations	American Association of Naturopathic Physicians: https://naturopathic.org/ accessed on 27 July 2022 American Nutrition Association: https://www.theana.org accessed on 27 July 2022 Institute for Functional Medicine: https://www.ifm.org/ accessed on 27 July 2022

**Table 7 nutrients-14-03934-t007:** A General Clinical Framework for Assessing Melatonin.

Clinical Aspect	Considerations
*Constitution*	
*Genes*, *Early Life Epigenetics*	Gene variants related to receptor activity, early life exposure to melatonin through breast milk
*Conditional Influences*	
*Acute and/or* *Chronic Triggers*	Stressful events, bouts of poor-quality sleep, travel across time zones, jet lag, inflammatory cytokines from injury or illness, oxidative stress from toxic exposures, shift work, dysregulated appetite, dysbiosis, artificial, blue light at night, insufficient darkness at night, insufficient morning light, highly processed, inflammatory diet
*Body Systems*	
*Bone Health*	Melatonin may help in the balance of osteoblasts and osteoclasts for better bone mineral density and overall structure.
*Brain Mood*	Melatonin can influence cognition and mood. High levels of kynurenine are present in the brain in depression.
*Cardiovascular Transport*	Melatonin can be produced in multiple body parts, circulate to tissues, and cross the blood–brain barrier.
*Detoxification*	Preliminary research suggests that melatonin may be helpful with the elimination of toxins in the brain (e.g., amyloid) through the glymphatic fluid.
*Endocrine System*	Melatonin is a hormone produced by the pineal gland (and gastrointestinal tract), communicating with other hormones. It is biochemically interrelated with its precursor, serotonin, and plays a key role in circadian rhythm and sleep cycles in conjunction with other hormones (e.g., cortisol, insulin) and neurotransmitters (e.g., serotonin). Higher amounts are found in children with lower nocturnal levels in puberty.
*Gastrointestinal Tract*	Melatonin is found in the gut mucosa at levels that exceed that of the pineal gland. It is produced by enterochromaffin cells, with altering responses postprandially. Furthermore, initial studies suggest it may influence the gut microbiome.
*Immunity*	Based on historical data, melatonin is most known as a potent antioxidant and anti-inflammatory agent. Research suggests it has chemopreventive and tumor-suppressing activity.
*Metabolism*	Melatonin can protect the mitochondria from oxidative stress due to its ability to cross the mitochondrial membrane.
*Lifestyle Factors*	
*Sleep Relaxation*	Aligning day-night rhythms will help to ensure healthy melatonin levels. Ensuring sleep hygiene is practiced, particularly maintaining a dark, cool room for sleeping. Wearing blue-light-blocking glasses before bedtime may help to establish better rhythm tone, enhanced sleep, and less reduction in nocturnal melatonin.
*Physical Activity*	Exercise may help increase serotonin and melatonin and result in less shunting through the kynurenine pathway.
*Nutrition*	Melatonin is found in both animal and plant dietary sources. Foods containing tryptophan may modulate melatonin levels due to the conversion of tryptophan to melatonin. Dietary supplementation could also be implemented either acutely, such as in jet lag, or more chronically at lower doses for those who do shift work.
*Stress Regulation * *Resilience*	Cortisol is inversely related to melatonin. Upregulation in the kynurenine pathway can be seen in stressful events. The use of meditation, calming activities, bodywork, and creative arts may help cultivate improved stress response and, ultimately, resilience.

## Data Availability

Not applicable.

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
