# Peer review of "Is Melatonin the “Next Vitamin D”?: A Review of Emerging Science, Clinical Uses, Safety, and Dietary Supplements"

_nutrients, 2022, doi:10.3390/nu14193934_

Round 1
Reviewer 1 Report
Very interesting paper in clinical point of view. Well presented every stage of publication. Especially worth is wide literature and discussion.
It should be underline only possible intraction between Viatmin D, glutations and melatonin .
It is well known glutation deficiency induces epigenetic alternation of viatmin D metablism genes incase of high-fat det-fed people. Is any correation in that sytuation?
Author Response
Dear Expert Reviewer,
Thank you kindly for your thoughtful comments. I will respond to them in blue font below.
Very interesting paper in clinical point of view. Well presented every stage of publication. Especially worth is wide literature and discussion.
Thank you very much for this gracious acknowledgment.
It should be underline only possible intraction between Viatmin D, glutations and melatonin .
We thank you for bringing this point to our attention. In fact, I published a review article on nutritional support for glutathione in Nutrients in 2019, and due to your astute comment, I was able to bring this review and other studies into this discussion. A separate section on glutathione was added as per below.
It is well known glutation deficiency induces epigenetic alternation of viatmin D metablism genes incase of high-fat det-fed people. Is any correation in that sytuation?
Thank you for this comment regarding glutathione deficiency, vitamin D metabolism, and high-fat diets. Per your suggestion to explore the correlation, we went into the literature further and added a paragraph (below) citing a few more studies, notably one from Parsanathan and Jain that addressed this research specifically. We thank you for the suggestion.
4.7.2.4 Glutathione
Glutathione is one of the major antioxidant systems in the body and is susceptible to changes in redox status and nutrient supply [312]. Various research studies have confirmed that there are individual associations between melatonin and glutathione [162], and glutathione and vitamin D [313]. Thus, while not yet extensively researched, there could be an interrelationship between the collective antioxidant activities of glutathione, vitamin D, and melatonin. Parsanathan and Jain [313] have demonstrated the effects of glutathione deficiency in high-fat-fed mice with observed epigenetic changes in genes responsible for vitamin D metabolism. Type 2 diabetics may be particularly vulnerable to decreases in antioxidants and increases in inflammatory markers. One recent clinical trial in type 2 diabetics found that modest vitamin D supplementation helped to increase both vitamin D and glutathione levels while decreasing oxidative stress and inflammation [314]. More research is required to understand how glutathione levels are implicated with both melatonin and vitamin D.
Reviewer 2 Report
In the current study, the authors made an overall review on melatonin and provide valuable information for the researchers. Several issues should be addressed to further improve the quality of the review.
1. In Figure 1, the melatonin level should be “pg/ml” rather than “µg/ml”. This was 1,000,000 holds of difference. Please correct it.
2. As to the similarity of melatonin and vitamin D, melatonin also can bind to the vitamin D receptor to produce its physiological activities. This should be mentioned in the text and the related references should be cited.
3. It is well known that melatonin is synthesized in mitochondria and chloroplasts. Its major activities may occur at the level of mitochondria. As an overall review, this novel information should be discussed in detail as a section in the text to improve the quality of the review (Tan, D.-X. and Reiter, R.J. 2019. Mitochondria: the birth place, battle ground and the site of melatonin metabolism in cells. Melatonin Research. 2, 1 (Feb. 2019), 44-66. doi.10.32794/mr11250011).
4. In the “2.3. Kynurenine Pathway, Energy Regulation & Stress Response” it was mentioned that “Low levels of melatonin may trigger an upregulation in the kynurenine pathway,”. This may be not the case, and in majority of the condition, it is the upregulation in the kynurenine pathway that suppresses melatonin production. Please clarify this issue.
Author Response
Dear Expert Reviewer,
Thank you kindly for your thoughtful comments. I will respond to them in bold font below.
In the current study, the authors made an overall review on melatonin and provide valuable information for the researchers. Several issues should be addressed to further improve the quality of the review.
- In Figure 1, the melatonin level should be “pg/ml” rather than “µg/ml”. This was 1,000,000 holds of difference. Please correct it.
Thank you for pointing out this error/discrepancy. The figure has been changed.
- As to the similarity of melatonin and vitamin D, melatonin also can bind to the vitamin D receptor to produce its physiological activities. This should be mentioned in the text and the related references should be cited.
This is an excellent point and further substantiates the connection between vitamin D and melatonin. This added text was included in the introduction of the manuscript (along with supportive references not shown here):
"There may even be levels of crosstalk and overlap between them that have not yet been fully elucidated but might have clinical relevance. For example, it has been demonstrated that melatonin can bind several target proteins, including enzymes, receptors, pores, and transporters. Most relevant for the discussion of this paper is that it can bind the vitamin D receptor (VDR), resulting in an enhancement of vitamin D’s signaling effects and subsequent cellular activities."
- It is well known that melatonin is synthesized in mitochondria and chloroplasts. Its major activities may occur at the level of mitochondria. As an overall review, this novel information should be discussed in detail as a section in the text to improve the quality of the review (Tan, D.-X. and Reiter, R.J. 2019. Mitochondria: the birth place, battle ground and the site of melatonin metabolism in cells. Melatonin Research. 2, 1 (Feb. 2019), 44-66. doi.10.32794/mr11250011).
Thank you so much for directing me to this paper, which I had not seen originally. I included this pivotal work in a few sections:
#1: This section added to introduction along with references:
"It has been proposed that both vitamin D and melatonin orchestrate many of their functions, especially related to redox status, at the level of the mitochondria. Concurrent with the age-related depletion in levels of vitamin D and melatonin, there is mitochondrial dysfunction, which has implications in a variety of clinical conditions that present differently through the seasons with changing light exposure."
#2: A separate section added on the mitochondria
In addition, I included a separate section to fortify the discussion on mitochondria and melatonin, referencing many of Dr. Tan and Dr. Reiter’s excellent papers (references not shown here, but are included in the amended draft, included the requested reference above):
"2.2 The Central Role of the Mitochondria
Newer data indicate that the mitochondria are pivotal for several aspects of melatonin: its production, metabolism, and activity through receptors. Rather than respond to the signals of the light/dark cycle or the pineal gland, the mitochondria can induce the production of melatonin based on intracellular need. Levels of melatonin are known to be higher in mitochondria compared with blood levels, most like due to the greater antioxidant requirements with the copious amounts of free radicals generated through the electron transport chain. Melatonin assists in mitochondrial redox balance through its ability to reduce the superoxide anion molecule from the electron transport chain and to directly scavenge free radicals. In addition to these functions, melatonin facilitates mitochondrial function by encouraging healthy endogenous levels of antioxidant defense enzymes like superoxide dismutase.
Mitochondrial dysfunction is one of the mechanisms related to diseases of aging. It may be that declining melatonin levels with age and, thus, less protection of the mitochondria from oxidative stress may be contributors to preclinical changes and, ultimately, clinical symptoms. There may be the ability to offset some of the decline accompanying the aging process through physiological regeneration with supplemental melatonin, as has been documented in animal and human studies."
- In the “2.3. Kynurenine Pathway, Energy Regulation & Stress Response” it was mentioned that “Low levels of melatonin may trigger an upregulation in the kynurenine pathway,”. This may be not the case, and in majority of the condition, it is the upregulation in the kynurenine pathway that suppresses melatonin production. Please clarify this issue.
Thank you for bringing up that important point. Additional clarification was added to that section along with references:
"Through this pathway and concurrent upregulation in inflammatory cytokines, melatonin and serotonin reserves may become depleted through suppressed production. Hence, there is some discussion that depression and/or anxiety can result through this upregulation and subsequent depletion of neurotransmitter substrates for healthy mood."
Thank you very much for these important comments to further strengthen the paper!